# *Drosophila* eIF3f1 mediates host immune defense by targeting dTak1

Yixuan Hu[1,2,6], Fanrui Kong[1,3,6], Huimin Guo [ID][1,4,6], Yongzhi Hua[1], Yangyang Zhu[1,3], Chuchu Zhang[1,3], Abdul Qadeer[1,3], Yihua Xiao[1,3], Qingshuang Cai [ID][5✉] & Shanming Ji [ID][1,3✉]

## Abstract

Eukaryotic translation initiation factors have long been recognized for their critical roles in governing the translation of coding RNAs into peptides/proteins. However, whether they harbor functional activities at the post-translational level remains poorly understood. Here, we demonstrate that *eIF3f1* (*eukaryotic translation initiation factor 3 subunit f1*), which encodes an archetypal deubiquitinase, is essential for the antimicrobial innate immune defense of *Drosophila melanogaster*. Our in vitro and in vivo evidence indicate that the immunological function of eIF3f1 is dependent on the N-terminal JAMM (JAB1/MPN/Mov34 metalloenzymes) domain. Mechanistically, eIF3f1 physically associates with dTak1 (*Drosophila* TGF-beta activating kinase 1), a key regulator of the IMD (immune deficiency) signaling pathway, and mediates the turnover of dTak1 by specifically restricting its K48-linked ubiquitination. Collectively, these results provide compelling insight into a noncanonical molecular function of a translation initiation factor that controls the post-translational modification of a target protein.

**Keywords** eIF3f1; dTak1; K48-linked Ubiquitination Modification; Antimicrobial Immune Defense; *Drosophila melanogaster*
**Subject Categories** Immunology; Post-translational Modifications & Proteolysis; Signal Transduction

## Introduction

Translational control mediates the efficiency of mRNAs into peptides/proteins and influences a wide range of cellular and physiological operations (Gebauer and Hentze, 2004; Hershey et al, 2012; Roux and Topisirovic, 2018; Song et al, 2021; Vogel and Marcotte, 2012). Evidence derived from eukaryotic systems substantiates that translational control is exerted at diverse junctures along the protein synthesis pathway, with the most frequently targeted phase, the rate-limiting initiation stage, modulated by an extensive array of eIFs (eukaryotic translation initiation factors) (Gebauer and Hentze, 2004; Jackson et al, 2010; Sonenberg and Hinnebusch, 2009; Vogel and Marcotte, 2012). As part of the translation initiation process, eIFs facilitate the recycling of dissociated ribosomal subunits, activate bound mRNAs, and recognize start codons, thereby orchestrating the assembly of an elongation-competent 80 S ribosome (Jackson et al, 2010; Pestova et al, 2001). Irregularities in eIFs' activities often incite aberrant protein expressions, resulting in potential cellular or tissue imbalances, or even systemic diseases (Sonenberg and Hinnebusch, 2009). Nevertheless, in spite of the pivotal roles that eIFs play in translational control, whether eIFs execute functional activities beyond translation remains poorly understood.

Innate immunity constitutes the host's first line of defense against the invasion of external pathogens (Beutler, 2004). It necessitates meticulous regulations, especially at the post-translational level (Liu et al, 2016). *Drosophila melanogaster* (fruit fly) provides an exemplary platform for investigating the regulatory mechanisms of the innate immune response, as it relies exclusively on innate immunity to neutralize microbial invaders, the manner of which is absent of the potential impacts of adaptive immunity (Hoffmann, 2003; Hultmark, 2003). In *Drosophila*, the humoral innate immune defense predominantly operates under the governance of several signaling cascades, for instance the Toll and the IMD (immune deficiency) pathways (Georgel et al, 2001; Hoffmann, 2003; Hultmark, 2003; Lemaitre et al, 1995; Lemaitre et al, 1996), the latter of which exhibits evolutionary similarities with the mammalian TNFR (tumor necrosis factor receptor) pathway (Kleino and Silverman, 2014; Myllymaki et al, 2014). Upon infection by some Gram-negative bacteria, the pattern recognition receptors upstream of the IMD pathway recognize the diaminopimelic acid-type peptidoglycan present in the bacterial cell wall (Guan et al, 2004; Werner et al, 2003; Werner et al, 2000), and instigate a series of downstream cascade reactions that lead to the phosphorylation, cleavage, and nuclear translocation of the NF-κB-like (nuclear factor-kappa B-like) protein Relish (Erturk-Hasdemir et al, 2009; Silverman et al, 2000; Stoven et al, 2000; Stoven et al, 2003). The nuclear-localized Relish therefore stimulates the expression of a diverse array of immune effectors, for instance AMPs (antimicrobial peptides) (Hoffmann, 2003; Hultmark, 2003; Georgel et al, 2001).

[1]Center for Developmental Biology, School of Life Sciences, Anhui Agricultural University, 230036 Hefei, Anhui, China. [2]Institutes of Brain Science, Wannan Medical College, 241002 Wuhu, Anhui, China. [3]Anhui Province Key Laboratory of Resource Insect Biology and Innovative Utilization, School of Life Sciences, Anhui Agricultural University, 230036 Hefei, Anhui, China. [4]Center for Biological Technology, Anhui Agricultural University, 230036 Hefei, Anhui, China. [5]Institut de Génétique et de Biologie Moléculaire et Cellulaire, Illkirch 67400, France. [6]These authors contributed equally: Yixuan Hu, Fanrui Kong, Huimin Guo. ✉E-mail: caiq@igbmc.fr; jism@ahau.edu.cn

The ubiquitination pathway, which represents one of the primary post-translational modifications of proteins, exerts a substantial influence on controlling the fly IMD innate immune response. It was demonstrated that Dredd (death-related ced-3/nedd2-like protein) is a key entity undergoing the K63 (63rd lysine)-linked ubiquitination, a process governed by the E3 ubiquitin ligase Diap2 (Drosophila IAP protein 2) (Kleino et al, 2005; Meinander et al, 2012). This E3 ligase also plays a critical role in facilitating the assembly of the K63-linked ubiquitination of the adaptor protein Imd (Paquette et al, 2010). In addition, many other E3 ligases, including Posh (plenty of SH3s) (Tsuda et al, 2005), Dnr1 (defense repressor 1) (Foley and O'Farrell, 2004; Guntermann et al, 2009), and SCF (skp-cullin-F-box) complex (Khush et al, 2002), have been posited to target dTak1 (Drosophila TGF-beta activating kinase 1), Dredd, and Relish, respectively, for ubiquitination. Conversely, an array of deubiquitinases (Dubs), including dUsp36 (Drosophila ubiquitin-specific protease 36) (Thevenon et al, 2009), Cyld (cylindromatosis) (Tsichritzis et al, 2007), Faf (fat facets) (Yagi et al, 2013), Usp2 (ubiquitin-specific protease 2) (Engel et al, 2014), dTrabid (Drosophila TRAF-binding domain-containing protein) (Fernando et al, 2014), and Otu (ovarian tumor) (Ji et al, 2019), has been identified as essential regulators in restricting the ubiquitination level of some key factors of the IMD pathway. These ubiquitin E3 ligases and Dubs together, orchestrate a critical ubiquitin-involved regulatory network, facilitating both the activation and the homeostatic maintenance of IMD signaling.

The Drosophila proteome comprises 41 cytoplasmic translation initiation factors (Marygold et al, 2017). Of major interest, the f1 subunit of the translation initiation factor 3, eIF3f1, harbors a JAMM (JAB1/MPN/Mov34 metalloenzymes) domain at the N-terminal region (Pahi et al, 2022; Tsou et al, 2012), which was thought to behave as a typically catalytical triad for deubiquitination reaction (Amerik and Hochstrasser, 2004). In this study, we delve into the functional role of eIF3f1 in regulating innate immunity in Drosophila. We show that silencing eIF3f1 hampers the fly IMD- but not Toll-mediated antimicrobial defense. Our molecular and biochemical analyses illuminate that eIF3f1 associates with dTak1 via the N-terminal JAMM domain, and specifically curtails dTak1's K48-linked ubiquitination modification, thereby bolstering the stability of dTak1. Our study not only unveils a noncanonical molecular role of eIF3f1 independent of translation initiation, but also broadens our insights into a previously uncharted regulatory mechanism by which dTak1 turnover is modulated by eIF3f1.

## Results

### eIF3f1 regulates IMD signaling in Drosophila S2 cells

We initially employed Drosophila S2 cells to perform a large-scale genetic screening by using the dsRNA-based RNAi (RNA interference) system. For this, S2 cells were first treated with various dsRNAs for 48 h (gfp dsRNA was used in the control group), followed by transfection with the Flag-Imd expression plasmid together with the Att-Luc (attacin promoter-luciferase) reporter system (Zhu et al, 2021), to monitor the relative level of IMD signaling. In this screening, we found that the Att-Luc level was decreased in the S2 cell sample treated with eIF3f1 dsRNA,

suggesting that eIF3f1 is a potential modulator in IMD signaling. Since the immune function of Drosophila eIF3f1 is largely unknown, we decided to investigate the functional involvement of eIF3f1 in modulating the IMD innate immunity in Drosophila.

To confirm our findings in the genetic screening, we synthesized three distinct dsRNAs that target either the coding region of eIF3f1 (referred to as eIF3f1 #1 and eIF3f1 #2), or the 3' untranslated region of eIF3f1 (referred to as eIF3f1 #3). The RNAi assay was performed by treating cultured S2 cells with these dsRNAs separately, or with control dsRNAs targeting the coding region of GFP or Rel (Relish), which encodes an essential transcription factor for IMD signaling. The knockdown efficiency of eIF3f1 dsRNAs in S2 cells was examined through Western blot assay (Fig. 1A, lower panel). These dsRNA-pretreated cells were further transfected with Flag-Imd and Att-Luc plasmids. As demonstrated in Fig. 1A, eIF3f1 silencing resulted in approximately a 60% reduction in the IMD signaling activities. Consistent results were obtained when RT-qPCR (reverse transcription plus quantitative polymerase chain reaction) assays were carried out to assess the transcript levels of several AMPs (antimicrobial peptides) downstream of the IMD pathway, including AttA (attacin A), Dpt (diptericin), and CecA1 (cecropin A1) (Fig. 1B–D). Of note, when we challenged S2 cells with heat-killed E. coli (Escherichia coli), we obtained similar results (Fig. EV1A–C). These findings suggest that eIF3f1 serves as a positive effector in the IMD pathway in Drosophila S2 cells.

To gain more insights into the regulatory role of eIF3f1 in IMD signaling, we constructed a plasmid expressing Myc-tagged eIF3f1 in S2 cells (Fig. 1E). As evidenced in Att-Luc (Fig. 1E) and RT-qPCR assays (Figs. 1F–H and EV1D–F), the overexpression of eIF3f1 amplified the Imd-induced Att-Luc activities and the corresponding AMP expressions in a measurement-dependent manner.

### eIF3f1 is dispensable for affecting Toll signaling

As mentioned in "Introduction", the Drosophila humoral immune responses are predominantly overseen by the Toll and the IMD signaling pathways (Hoffmann, 2003; Hultmark, 2003). We next probed the potential involvement of eIF3f1 in the regulation of the Toll pathway. By utilizing a previously established Toll$^{\Delta LRR}$-driven Drs-Luc (drosomycin promoter-luciferase) reporter system (Ji et al, 2014), we observed that neither the downregulation nor the overexpression of eIF3f1 exerted any significant impacts on the Drs-Luc activities (Fig. EV2A,D). In agreement with this, altering eIF3f1 expression resulted in negligible influences on the inductions of Toll downstream AMPs, including Drs (Drosomycin) and Mtk (Metchnikowin) (Fig. EV2B,C,E,F). Collectively, our results indicate that eIF3f1 fulfills a role in mediating the IMD but not the Toll signaling pathway in Drosophila S2 cells.

### eIF3f1 is required for host defense against microbial infection

We next explored the functional role of eIF3f1 in vivo. As the ubiquitous loss of eIF3f1 leads to lethality at the larval stage of Drosophila (Tsou et al, 2012), we therefore generated tissue-specific eIF3f1 silencing by crossing the eIF3f1 RNAi line with the c564-gal4 strain. The c564-gal4 controls gene expression mostly in the fat body, which is one of the main responsible tissues/organs during

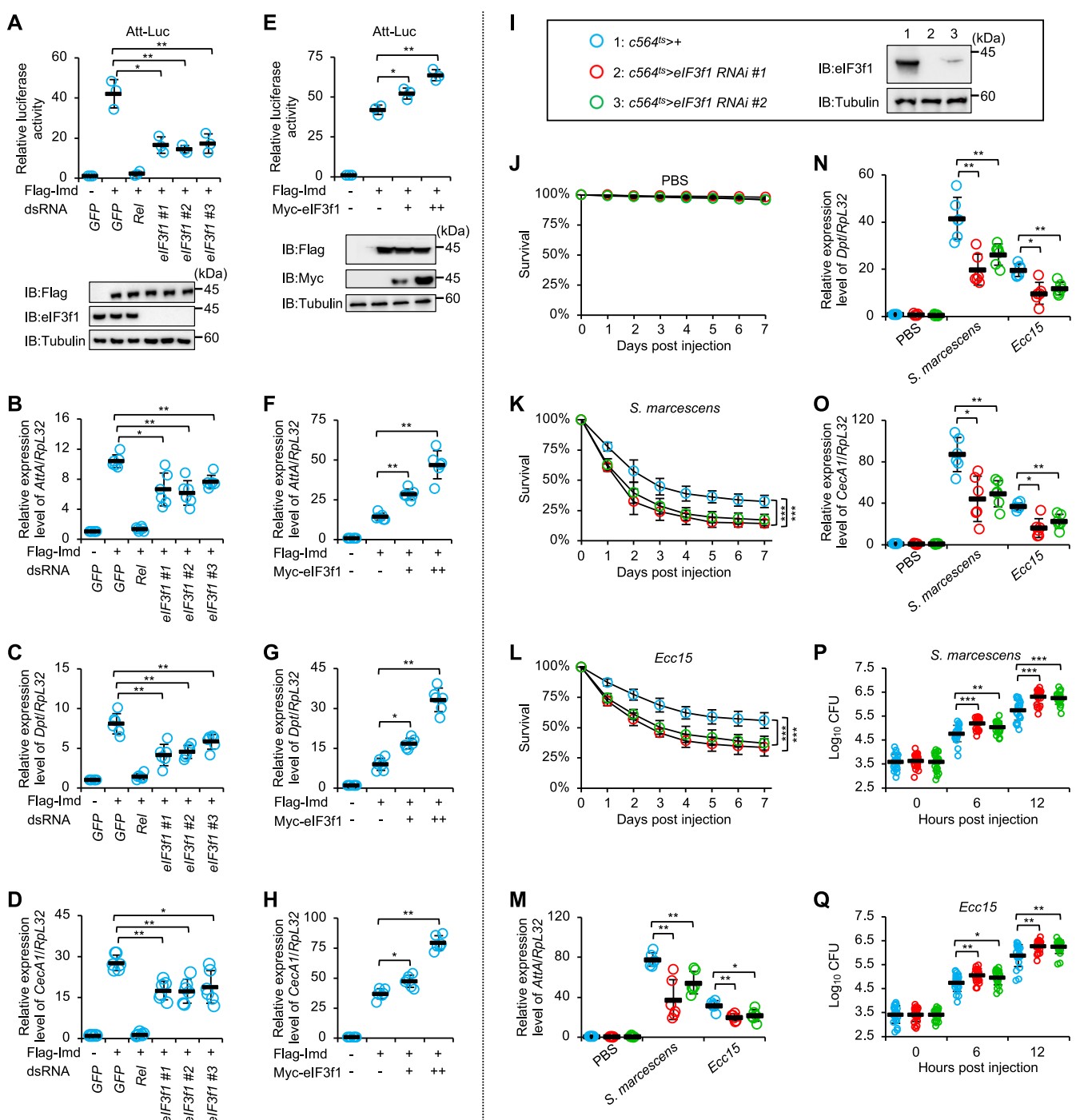

systemic infection of *Drosophila*. The *tub-gal80^ts* strain was employed to permit eIF3f1 silencing at the adult stage (Fig. 1I). We injected flies, including *c564^ts > eIF3f1 RNAi #1*, *c564^ts > eIF3f1 RNAi #2*, and *c564^ts > +* (*wild-type* control), with sterile PBS buffer (control treatment), Gram-negative bacteria *S. marcescens* (*Serratia marcescens*) or *Ecc15* (*Pectobacterium carotovorum carotovorum 15*), or Gram-positive bacterium *E. faecalis* (*Enterococcus faecalis*) for survival analyses. As illustrated in Fig. 1J–L, the *eIF3f1 RNAi* flies succumbed much faster than the control flies upon infection of *S. marcescens* or *Ecc15*. However, all flies showed similar survival

rates after *E. faecalis* stimuli (Fig. EV3A). We then evaluated the inductions of several AMPs in these flies following bacterial challenge. The transcript levels of *AttA*, *Dpt*, and *CecA1* in the *eIF3f1 RNAi* flies were lower (with a reduction of ~30–60%) than those in the control group post-injection of *S. marcescens* or *Ecc15* (Figs. 1M–O and EV3B). Notably, we did not observe significant differences in these flies concerning the inductions of *Drs* or *Mtk* in response to *E. faecalis* infection (Fig. EV3C,D).

To determine whether eIF3f1 is involved in modulating the proliferation of injected bacteria, we performed a time-course

**Figure 1.   eIF3f1 plays a critical role in the fly antimicrobial immune defenses.**

(A) *Drosophila* S2 cells were treated with various dsRNAs (3 μg per dsRNA) separately. Forty-eight hours later, cells were transfected with Imd expressing plasmid (0.5 μg) or empty vector (0.5 μg), together with Att-Luc (0.5 μg) and Renilla (0.1 μg) plasmids. 36 h post transfection, cells were harvested for Att-Luc assays (upper panel). Western blot experiments were performed to monitor the expression levels of different proteins (lower panel). Tubulin was used as loading control. (B–D) S2 cells were treated with indicated dsRNAs (3 μg per dsRNA) for 48 h. Cells were then transfected with Imd expressing plasmid (0.5 μg) or empty vector (0.5 μg) for 36 h, and subjected to RT-qPCR assays to examine the transcript levels of *AttA* (B), *Dpt* (C), and *CecA1* (D). (E) S2 cells were transfected with Imd (0.5 μg) and eIF3f1 (0, 0.5, and 1 μg) expressing plasmids, together with Att-Luc (0.5 μg) and Renilla (0.1 μg) as indicated. Thirty-six hours post transfection, cells were harvested for Att-Luc assays. (F–H) S2 cells were transfected with Imd (0.5 μg) and eIF3f1 (0, 0.5, and 1 μg) expressing plasmids for 36 h. Cells were then harvested for RT-qPCR analyses. (I) Fat body samples were dissected from adult flies including *c564^{ts} > +*, *c564^{ts} > eIF3f1 RNAi #1*, and *c564^{ts} > eIF3f1 RNAi #2*, respectively. Samples were then subjected to Western blot assays to examine the RNAi efficiency of *eIF3f1*. (J–L) Flies including *c564^{ts} > +*, *c564^{ts} > eIF3f1 RNAi #1*, and *c564^{ts} > eIF3f1 RNAi #2* were injected with PBS (J), *S. marcescens* (K), or *Ecc15* (L), and subjected to survival analyses. The numbers of flies are as follows. (J) *c564^{ts} > +*: 88, 86, 85; *c564^{ts} > eIF3f1 RNAi #1*: 84, 82, 84; *c564^{ts} > eIF3f1 RNAi #2*: 89, 88, 87. (K) *c564^{ts} > +*: 87, 84, 93; *c564^{ts} > eIF3f1 RNAi #1*: 83, 82, 88; *c564^{ts} > eIF3f1 RNAi #2*: 86, 87, 89. (L) *c564^{ts} > +*: 87, 92, 88; *c564^{ts} > eIF3f1 RNAi #1*: 90, 84, 85; *c564^{ts} > eIF3f1 RNAi #2*: 89, 85, 87. (M–O) Flies including *c564^{ts} > +*, *c564^{ts} > eIF3f1 RNAi #1*, and *c564^{ts} > eIF3f1 RNAi #2* were injected with PBS, *S. marcescens*, or *Ecc15*. Six hours later, flies were harvested for RT-qPCR assays. (P, Q) Flies including *c564^{ts} > +*, *c564^{ts} > eIF3f1 RNAi #1*, and *c564^{ts} > eIF3f1 RNAi #2* were injected with *S. marcescens* or *Ecc15*. At indicated time points, flies were collected for bacterial burden analyses. Data Information: (A–H, M–Q) Each dot represents one biological replicate. Data are shown as means ± SD (standard errors). The nonparametric Kruskal–Wallis test was used for statistical analyses. (K, L) The Log-Rank test was used for statistical analyses. *$P < 0.05$; **$P < 0.01$; ***$P < 0.001$. Source data are available online for this figure.

(0, 6, and 12 h post-injection) bacterial burden assay. We detected an increase (by ~50–200%) in the colony-forming units of both *S. marcescens* and *Ecc15* in the *eIF3f1 RNAi* flies, compared to those in the controls (Fig. 1P,Q). However, we hardly found marked alterations in the *E. faecalis* population among all samples (Fig. EV3E). Taken together, our data clearly demonstrate that eIF3f1 plays a role in governing the fly IMD innate immune defense against bacterial infection.

## eIF3f1 regulates IMD innate immunity in a JAMM domain-dependent manner

eIF3f1 harbors a JAMM (JAB1/MPN/Mov34 metalloenzymes) domain at its N-terminal region (amino acids 1–137) (Pahi et al, 2022; Tsou et al, 2012). To delineate the functional domain required for eIF3f1's role in innate immunity, we constructed plasmids expressing two different truncated forms of eIF3f1, including eIF3f1^{JAMM} and eIF3f1^{ΔJAMM}, in S2 cells (Fig. 2A). By again utilizing the Att-Luc reporter system, we discerned that eIF3f1^{JAMM} mimicked the effect seen with eIF3f1^{FL}, but this was not the case for eIF3f1^{ΔJAMM} (Fig. 2B). This finding was further confirmed in RT-qPCR analyses (Fig. 2C–E). Of major interest, when we conducted rescue experiments, we observed that the overexpression of eIF3f1^{FL} or eIF3f1^{JAMM}, but not eIF3f1^{ΔJAMM}, effectively counteracted the downregulation of IMD signaling by silencing eIF3f1 (Fig. 2F–I). These results indicate that the JAMM domain of eIF3f1 is both necessary and sufficient for eIF3f1 controlling IMD signaling.

To corroborate these findings in vivo, we generated several transgenic fly strains and crossed them with the *c564-gal4* line to obtain progenies with ectopic expression of eIF3f1^{FL}, eIF3f1^{JAMM}, or eIF3f1^{ΔJAMM} in the fat body. Survival analyses following microbial infections displayed that the overexpression of eIF3f1^{FL} or eIF3f1^{JAMM} in the fat body reduced the fly mortality upon *S. marcescens* or *Ecc15* challenge (Fig. 2J–L). Moreover, ectopic eIF3f1^{FL} or eIF3f1^{JAMM} led to an augmentation (over 40%) of the AMP inductions (Fig. 2M–O) and a reduction (over 30%) of the bacterial burdens (Fig. 2P,Q).

Next, we carried out genetic crossings using these transgenic flies for rescue experiments. As illustrated in Fig. 3A–C, the survival defects of the *eIF3f1 RNAi* flies after microbial infections were ameliorated by the overexpression of eIF3f1^{FL} or eIF3f1^{JAMM}.

Consistently, the aberrant AMP inductions (Fig. 3D–F) and increased bacterial proliferation (Fig. 3G,H) in the *eIF3f1 RNAi* flies were reversed by the overexpression of eIF3f1^{FL} or eIF3f1^{JAMM}, but not eIF3f1^{ΔJAMM}.

In summary, our findings demonstrate that the JAMM domain plays a critical role in eIF3f1's positive influence on the fly antimicrobial defense.

## eIF3f1 mediates dTak1 levels without affecting *dTak1* translation

We further elucidated the molecular mechanism underpinning the role of eIF3f1 in modulating *Drosophila* innate immunity. As *Drosophila* eIF3f1 is potentially an integral component of the translation initiation complex, we posited that eIF3f1 probably control the translation of some key factor(s) of the core IMD pathway. We performed the puromycin-involved translation examination assay (Schmidt et al, 2009), and observed that the global protein synthesis was enhanced by the overexpression of eIF3f1 in S2 cells (Fig. 4A). To pinpoint the protein candidate(s) regulated by eIF3f1, we conducted an LC-MS/MS (liquid chromatography-tandem mass spectrometry) analysis, in which more than 1000 candidates were identified. We found that the overexpression of eIF3f1 increased the levels of 276 proteins/peptides, even though the majority of the detected candidates were not significantly affected (Fig. 4B). Gene ontology analysis of these upregulated proteins/peptides revealed that they primarily function in the processes of "glycosylation", "biosynthesis", and "translation" (Fig. 4C), implying that the overexpression of eIF3f1 indeed affects the protein translation and synthesis in S2 cells. Interestingly, we noted an elevation in the protein level of dTak1 (*Drosophila* TGF-beta activating kinase 1), a pivotal kinase in the IMD signaling pathway (Silverman et al, 2003), in response to eIF3f1 over-expression (Fig. 4D). This was further corroborated by a western blot analysis in S2 cells (Fig. 4E,F).

To substantiate the impact of eIF3f1 on the translation of *dTak1* mRNA into protein, we initially isolated polysomes by utilizing the size exclusion chromatography, where heavy polysomes eluted earlier than light polysomes, with monosomes in the end (Fig. 4G). Consistent with the above findings, the overexpression of eIF3f1 resulted in an increase in the polysome peak (Fig. 4G). Transcripts accumulated in the heavier fractions are transcribed more

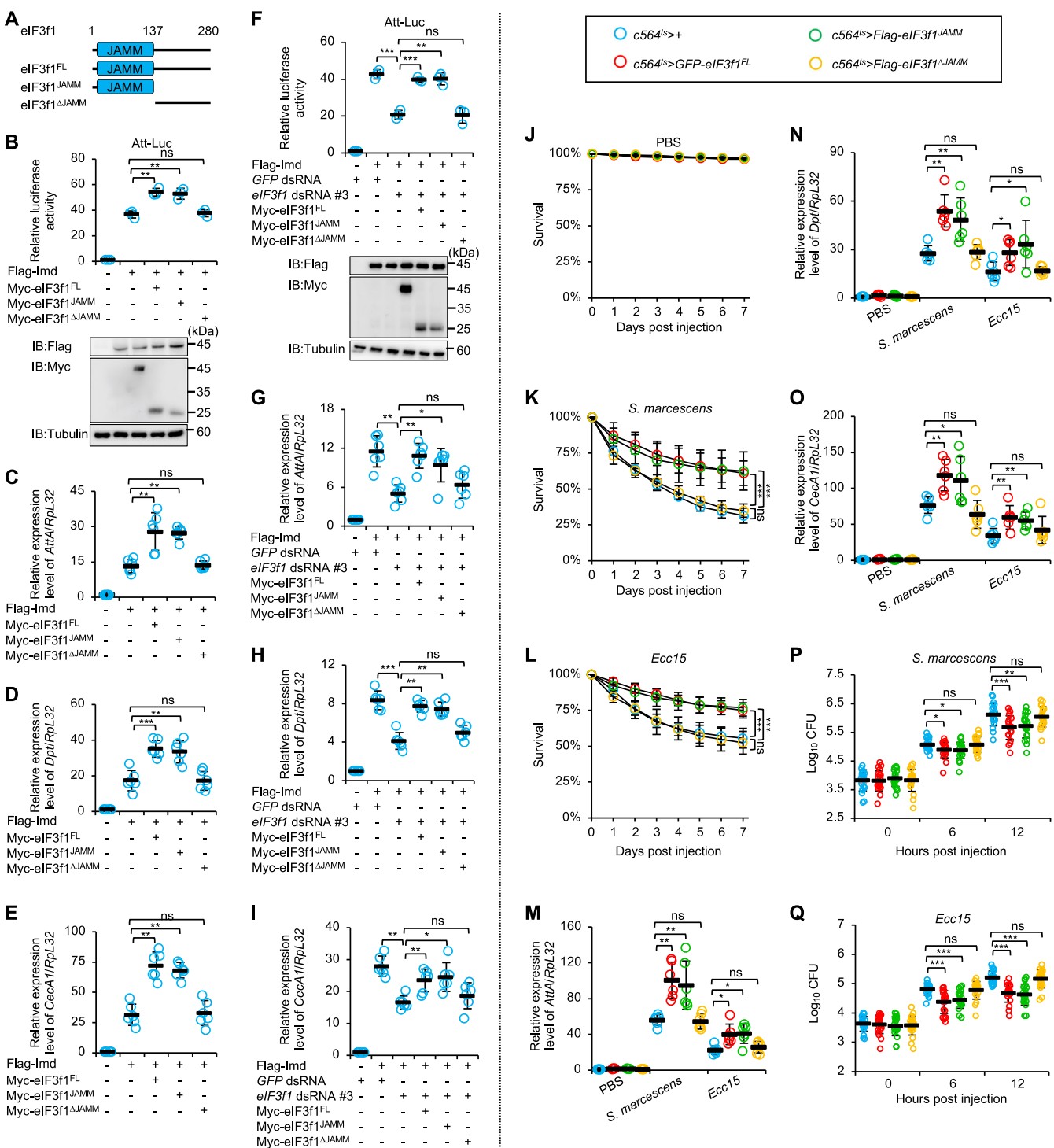

efficiently, as there are more ribosomes per transcript in the heavier fractions than those in the lighter fractions. We subsequently extracted total RNAs from various polysome and monosome fractions and performed RT-qPCR to monitor the *dTak1* mRNA contents. Contrary to our expectations, we did not observe significant alterations in the levels of *dTak1* transcripts within these purified polysomes or monosomes (Fig. 4H). These observations indicate that eIF3f1 may not essentially influence *dTak1* translation.

## eIF3f1 mediates dTak1 ubiquitination

Given that eIF3f1 fulfills its contribution to IMD signaling depending on the JAMM domain (Figs. 2 and 3), which was thought to be a catalytical triad for deubiquitination (Amerik and Hochstrasser, 2004), we hypothesized that eIF3f1 regulates dTak1 levels in a ubiquitin-involved manner. To determine whether eIF3f1 associates with dTak1, we conducted a co-immunoprecipitation assay in cultured S2 cells. We observed a

Figure 2.   The N-terminal JAMM domain is required for eIF3f1 mediating *Drosophila* innate immunity.

(A) Domain architecture of eIF3f1. (B) S2 cells were transfected with indicated plasmids (0.5 μg for each plasmid), together with Att-Luc (0.5 μg) and Renilla (0.1 μg) plasmids. Thirty-six hours later, cells were harvested for Att-Luc assays. (C–E) S2 cells were transfected with various plasmids (0.5 μg for each plasmid) for 36 h and then subjected to RT-qPCR assays. (F) S2 cells were treated with indicated dsRNAs (3 μg for each dsRNA) for 48 h. Cells were then transfected with different combinations of plasmids (0.5 μg for each plasmid), together with Att-Luc (0.5 μg) and Renilla (0.1 μg) plasmids for 36 h, and subjected to Att-Luc assays. (G-I) S2 cells were treated with different dsRNAs (3 μg for each dsRNA) for 48 h, transfected with various plasmids (0.5 μg for each plasmid) for 36 h as indicated, and subjected to RT-qPCR assays. (J–L) Flies including $c564^{ts} > +$, $c564^{ts} > GFP\text{-}eIF3f1^{FL}$, $c564^{ts} > Flag\text{-}eIF3f1^{JAMM}$, and $c564^{ts} > Flag\text{-}eIF3f1^{\Delta JAMM}$ were injected with PBS (J), *S. marcescens* (K), or *Ecc15* (L), followed by survival analyses. The numbers of flies are as follows. (J) $c564^{ts} > +$: 93, 91, 90; $c564^{ts} > GFP\text{-}eIF3f1^{FL}$: 94, 88, 93; $c564^{ts} > Flag\text{-}eIF3f1^{JAMM}$: 86, 92, 89; $c564^{ts} > Flag\text{-}eIF3f1^{\Delta JAMM}$: 89, 92, 87. (K) $c564^{ts} > +$: 86, 89, 92; $c564^{ts} > GFP\text{-}eIF3f1^{FL}$: 89, 91, 94; $c564^{ts} > Flag\text{-}eIF3f1^{JAMM}$: 87, 88, 89; $c564^{ts} > Flag\text{-}eIF3f1^{\Delta JAMM}$: 91, 84, 87. (L) $c564^{ts} > +$: 84, 89, 92; $c564^{ts} > GFP\text{-}eIF3f1^{FL}$: 89, 91, 93; $c564^{ts} > Flag\text{-}eIF3f1^{JAMM}$: 85, 91, 89; $c564^{ts} > Flag\text{-}eIF3f1^{\Delta JAMM}$: 91, 86, 90. (M-O) Flies including $c564^{ts} > +$, $c564^{ts} > GFP\text{-}eIF3f1^{FL}$, $c564^{ts} > Flag\text{-}eIF3f1^{JAMM}$, and $c564^{ts} > Flag\text{-}eIF3f1^{\Delta JAMM}$ were injected with PBS, *S. marcescens*, or *Ecc15*. 6 h later, flies were harvested for RT-qPCR assays. (P, Q) Flies including $c564^{ts} > +$, $c564^{ts} > GFP\text{-}eIF3f1^{FL}$, $c564^{ts} > Flag\text{-}eIF3f1^{JAMM}$, and $c564^{ts} > Flag\text{-}eIF3f1^{\Delta JAMM}$ were injected with *S. marcescens* or *Ecc15*. At indicated time points, flies were collected for bacterial burden analyses. Data Information: (B-I, M-Q) Each dot represents one biological replicate. Data are shown as means ± SD. The nonparametric Kruskal–Wallis test was used for statistical analyses. (K, L) The Log-Rank test was used for statistical analyses. *$P < 0.05$; **$P < 0.01$; ***$P < 0.001$; ns not significant, $P > 0.05$. Source data are available online for this figure.

substantial interaction between eIF3f1 and dTak1, a phenomenon that was not mirrored in the case of Imd under the same experimental condition (Fig. 5A). A domain mapping analysis further underscored that the JAMM domain is a prerequisite for eIF3f1 associating with dTak1 (Fig. 5B).

Despite that eIF3f1 was suggested to function in the deubiquitination of targeted proteins (Pahi et al, 2022; Tsou et al, 2012), empirical evidence substantiating its Dub enzymatic activity is lacking. Intriguingly, our in vitro deubiquitination assay explicitly ascertained this aspect (Fig. 5C). We then examined whether eIF3f1 affects the ubiquitination pattern of dTak1 in S2 cells. As shown in Fig. 5D,E, the overexpression of eIF3f1$^{FL}$, but not eIF3f1$^{\Delta JAMM}$, led to a reduction (over 70%) in the ubiquitination level of dTak1. Importantly, these findings were further confirmed in the fat body (Fig. 5F,G).

Recent studies by Fernando et al (Fernando et al, 2014) and us (Hua et al, 2022) underscored the critical role of both the K48 (48th lysine) and the K63 (63rd lysine)-linked ubiquitination of dTak1 in modulating IMD signaling. We therefore leveraged plasmids that express two types of ubiquitin mutants, UbK48 and UbK63, in which all lysines in ubiquitin were modified to arginines with the exception of K48 and K63, respectively. As demonstrated in ubiquitination assays, the overexpression of eIF3f1$^{FL}$, but not eIF3f1$^{\Delta JAMM}$, resulted in a reduction of the K48-linked ubiquitination of dTak1 (Fig. 5H,I). However, no significant changes were observed in the levels of the K63-linked ubiquitination of dTak1 among all samples (Fig. EV4A,B). Further, our ex vivo and in vivo studies displayed that the downregulation of eIF3f1 enhanced the K48-linked ubiquitination of dTak1 (Fig. 5J–M). Put together, these results indicate that eIF3f1, through its JAMM domain, associates with dTak1 and constrains the K48-linked ubiquitination modification of dTak1.

## eIF3f1 regulates dTak1 turnover

Considering that proteins normally undergo proteasomal degradation upon K48-linked ubiquitination, we sought to ascertain whether eIF3f1-mediated dTak1 deubiquitination is related to dTak1 stability. To this end, we transfected the Flag-dTak1 and Flag-Renilla plasmids into S2 cells that were pretreated with the *eIF3f1* or *gfp* (control) dsRNAs. Our western blot assay demonstrated a reduction (over 50%) in the protein level of dTak1 by the knockdown of eIF3f1, an effect that was not observed in the case of Renilla (Fig. 6A,B). This phenomenon was also observed in the

ubiquitination assay in S2 cells (Fig. EV4C). Moreover, the reduction of dTak1 levels was markedly reversed upon treatment with MG132, a widely used proteasome inhibitor (Fig. 6A,B). These data suggest that eIF3f1 regulates the proteasome-dependent degradation of dTak1 in S2 cells. We further performed the protein stability assay and found that silencing eIF3f1 accelerated the turnover of dTak1 (Fig. 6C,D). Consistently, the downregulation of eIF3f1 reduced the protein level of endogenous dTak1 in both S2 cells and fat body cells (Fig. EV4D,E).

To further elucidate the functional relevance of eIF3f1 in modulating dTak1 stability, we employed plasmids expressing eIF3f1$^{FL}$ and eIF3f1$^{\Delta JAMM}$. Our results, as depicted in Fig. 6E–H, showed that the overexpression of eIF3f1$^{FL}$ inhibited the degradation of dTak1, resulting in an elevated protein level of dTak1. Conversely, in the absence of the JAMM domain, eIF3f1 displayed a negligible impact on the stability of dTak1. Our findings underscore that eIF3f1 operates as a Dub and restricts the K48-linked ubiquitination modification of dTak1, thereby antagonizing the proteasome-mediated degradation of dTak1.

### eIF3f1 genetically interacts with *dTak1*

To investigate if eIF3f1 modulates *Drosophila* innate immunity via dTak1 in vivo, we employed the $c564^{ts} > dTak1\ RNAi;GFP\text{-}eIF3f1$ flies for bacterial infections. Silencing dTak1 almost entirely negated the elevated AMP expressions by the overexpression of eIF3f1 (Fig. 6I±K). Moreover, the heightened resistances to both *S. marcescens* and *Ecc15* observed in $c564^{ts} > GFP\text{-}eIF3f1$ flies were notably mitigated by *dTak1 RNAi* (Fig. 6L–P). Collectively, our results support the proposition that eIF3f1 modulates the host antimicrobial immune defense largely in a dTak1-dependent manner.

## Discussion

Translation initiation factors have been firmly established as essential players in translational control (Jackson et al, 2010; Sonenberg and Hinnebusch, 2009). Nevertheless, our knowledge with regard to the potentially functional activities of these factors beyond translation is lacking. In the present study, we uncover an unprecedented function of eIF3f1, the *Drosophila* ortholog of mammalian translation initiation factor eIF3F. We provide compelling evidence showing that eIF3f1 mediates the protein

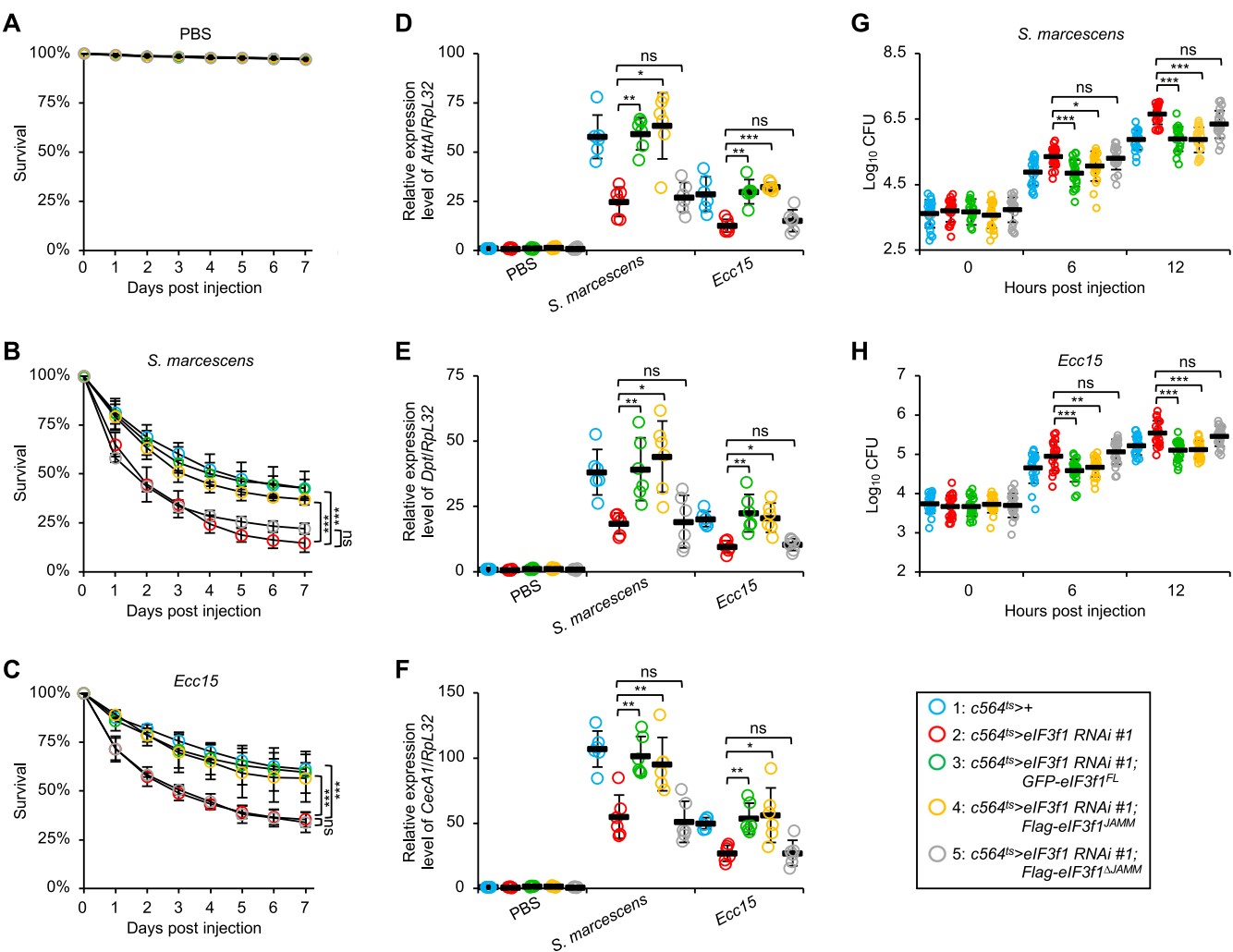

**Figure 3.** Overexpression of *eIF3f1* rescues the immune defects in *eIF3f1 RNAi* flies upon microbial infections.

(A–C) Flies including *c564^ts^ > +*, *c564^ts^ > eIF3f1 RNAi #1*, *c564^ts^ > eIF3f1 RNAi #1;GFP-eIF3f1^FL^*, *c564^ts^ > eIF3f1 RNAi #1;Flag-eIF3f1^JAMM^*, and *c564^ts^ > eIF3f1 RNAi #1;Flag-eIF3f1^ΔJAMM^* were injected with PBS (A), *S. marcescens* (B), or *Ecc15* (C), followed by survival analyses. The numbers of flies are as follows. (A) *c564^ts^ > +*: 95, 88, 91; *c564^ts^ > eIF3f1 RNAi #1*: 92, 89, 85; *c564^ts^ > eIF3f1 RNAi #1;GFP-eIF3f1^FL^*: 89, 87, 88; *c564^ts^ > eIF3f1 RNAi #1;Flag-eIF3f1^JAMM^*: 87, 89, 95; *c564^ts^ > eIF3f1 RNAi #1;Flag-eIF3f1^ΔJAMM^*: 95, 90, 89. (B) *c564^ts^ > +*: 89, 95, 88; *c564^ts^ > eIF3f1 RNAi #1*: 87, 86, 95; *c564^ts^ > eIF3f1 RNAi #1;GFP-eIF3f1^FL^*: 86, 93, 89; *c564^ts^ > eIF3f1 RNAi #1;Flag-eIF3f1^JAMM^*: 93, 89, 88; *c564^ts^ > eIF3f1 RNAi #1;Flag-eIF3f1^ΔJAMM^*: 88, 89, 86. (C) *c564^ts^ > +*: 87, 90, 93; *c564^ts^ > eIF3f1 RNAi #1*: 92, 89, 87; *c564^ts^ > eIF3f1 RNAi #1;GFP-eIF3f1^FL^*: 90, 91, 91; *c564^ts^ > eIF3f1 RNAi #1;Flag-eIF3f1^JAMM^*: 91, 89, 87; *c564^ts^ > eIF3f1 RNAi #1;Flag-eIF3f1^ΔJAMM^*: 86, 88, 89. (D–F) Flies including *c564^ts^ > +*, *c564^ts^ > eIF3f1 RNAi #1*, *c564^ts^ > eIF3f1 RNAi #1;GFP-eIF3f1^FL^*, *c564^ts^ > eIF3f1 RNAi #1;Flag-eIF3f1^JAMM^*, and *c564^ts^ > eIF3f1 RNAi #1;Flag-eIF3f1^ΔJAMM^* were injected with PBS, *S. marcescens*, or *Ecc15*. 6 h later, flies were harvested for RT-qPCR assays. (G, H) Flies including *c564^ts^ > +*, *c564^ts^ > eIF3f1 RNAi #1*, *c564^ts^ > eIF3f1 RNAi #1;GFP-eIF3f1^FL^*, *c564^ts^ > eIF3f1 RNAi #1;Flag-eIF3f1^JAMM^*, and *c564^ts^ > eIF3f1 RNAi #1;Flag-eIF3f1^ΔJAMM^* were injected with *S. marcescens* or *Ecc15*. At indicated time points, flies were collected for bacterial burden analyses. Data Information: (B, C) The Log-Rank test was used for statistical analyses. (D–H) Each dot represents one biological replicate. Data are shown as means ± SD. The nonparametric Kruskal–Wallis test was used for statistical analyses. $*P < 0.05$; $**P < 0.01$; $***P < 0.001$; ns not significant, $P > 0.05$. Source data are available online for this figure.

level of dTak1 through a post-translational but not translational control of dTak1, thereby positively contributing to the host antimicrobial immune defense. Our data shed lights on not only a noncanonical role of eIF3f1 but also an undescribed regulatory manner of the turnover of dTak1.

## Translation initiation factor functions at the post-translational level

To date, more than 40 distinct eIFs have been identified in eukaryotic organisms, with variations in the constituent subunits and functional roles across different species (Gebauer and Hentze, 2004; Sonenberg and Hinnebusch, 2009; Vogel and Marcotte, 2012). Intriguingly, our experimental approaches in *Drosophila* S2 cells demonstrate a functional requirement of eIF3f1 in the regulation of the IMD innate immune signaling pathway. Detailed genetic investigations further ascertain the crucial role of eIF3f1 in the fly immune response against bacterial infections. In fact, the immunological role of eIF3f1 appears to largely depend on dTak1, as evidenced by: (1) eIF3f1 antagonizes the K48-linked ubiquitination and proteosome-involved degradation of dTak1; and (2) silencing dTak1 almost fully abolishes the critical impact of eIF3f1 on *Drosophila* innate immunity. Initially, we

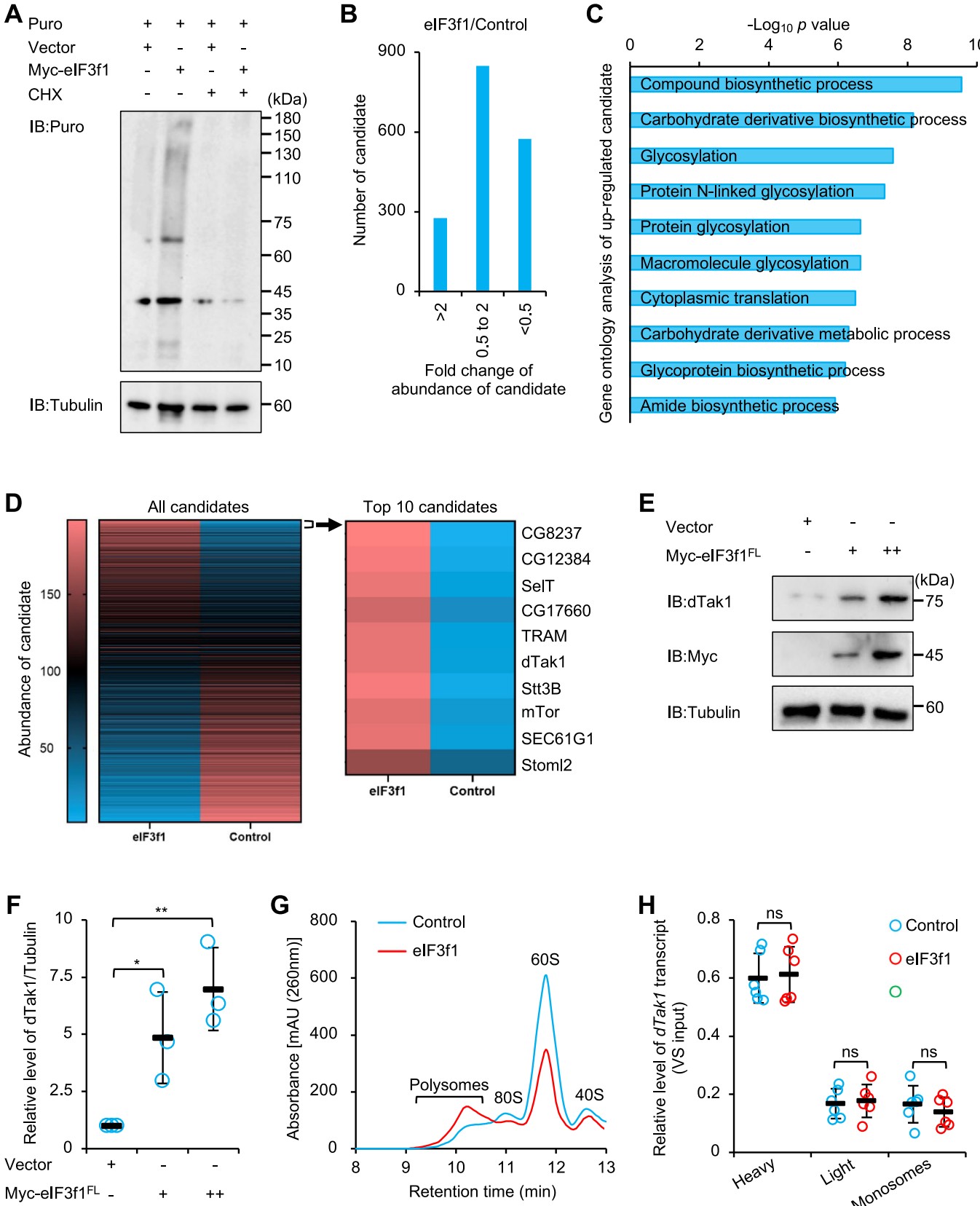

Figure 4.  eIF3f1 modulates dTak1 levels in S2 cells without affecting *dTak1* translation.

(A) S2 cells were transfected with eIF3f1 or empty plasmid for 36 h later. Cells were then treated with CHX (50 μg/ml) or DMSO (control) for 6 h as indicated, and subjected to translation efficiency assay. (B–D) S2 cells were transfected with eIF3f1 or empty plasmid for 36 h later. Cells were harvested for LC-MS/MS assay, and the abundances of detected peptides/proteins (candidates) were analyzed (eIF3f1 versus Control). The numbers of candidates in different categories are shown in (B). The GO analyses of upregulated candidates are shown in (C). All candidates and top ten upregulated candidates are shown in (D). (E, F) S2 cells were transfected with eIF3f1 (0.5 and 1 μg) or empty (0.5 μg) plasmids. 36 h later, cells were lysed for Western blot assays. Quantification of indicated protein levels are shown in (F). (G, H) S2 cells were transfected with eIF3f1 (0.5 μg) or empty (0.5 μg) plasmids for 36 h. Samples were then subjected to polysome fractionation by size exclusion chromatography and the polysomal profiles (at 260 nm) are shown in (G). RT-qPCR assays were performed to monitor the *dTak1* mRNA levels in different polysomes and monosomes (H). Data Information: (F, H) Each dot represents one biological replicate. Data are shown as means ± SD. The nonparametric Kruskal–Wallis test was used for statistical analyses. *$P < 0.05$; **$P < 0.01$; ns not significant, $P > 0.05$. Source data are available online for this figure

speculated that as a potential part of the translation initiation factors, eIF3f1 might directly modulate *dTak1* translation. However, our studies in S2 cells display that the overexpression of eIF3f1 hardly affects the translation efficiency of *dTak1*, despite that it results in a global enhancement in protein translation. As these experiments were carried out in cultured S2 cells with the overexpression of eIF3f1, the functional involvement of *Drosophila* eIF3f1 in translation initiation needs to be further investigated in more details. Previous researches suggested the potential Dub activity of eIF3f1, due to the presence of a JAMM domain at its N-terminus (Pahi et al, 2022; Tsou et al, 2012). Indeed, our biochemical evidence clearly illustrate that eIF3f1 relies on the JAMM domain to associate with dTak1, thereby preventing the K48-linked ubiquitin assembly and turnover of dTak1. Our studies delineate a functional role of eIF3f1 in post-translational regulation, a departure from its traditionally understood role in translational control.

Besides *eIF3f1*, the *Drosophila eIF3f* gene family also includes *eIF3f2*. In fact, both eIF3f1 and eIF3f2 harbor a JAMM domain and share 48% and 37% similarities with the mammalian eIF3F (Marygold et al, 2017), which was suggested to regulate the Notch signaling pathway via relying on its intrinsic Dub activity (Moretti et al, 2010). These findings raise a possibility that eIF3f2 plays a role similar to that of eIF3f1. To investigate the potential involvement of eIF3f2 in *Drosophila* innate immunity, we subjected both the *eIF3f2 RNAi* and the *eIF3f2 overexpression* flies to bacterial infections. We observed that neither the downregulation nor the overexpression of eIF3f2 affected the fly survival, the AMP expressions, or the bacterial burdens upon microbial infection (Fig. EV5A–F). Regardless, it is not possible to formally exclude the possibility that the translation initiation function might be performed by eIF3f2. A series of recent studies have demonstrated that some of the fly eIF3 subunits, for instance eIF3d, eIF3m, and eIF3j, are involved in regulating tissue/organ development, stress response, and so on (Ameijeiras et al, 2023; Ramat et al, 2020; Song et al, 2022; Szostak et al, 2018). All these functions are fulfilled through a translational control-dependent manner. It would therefore be worthwhile to explore the potential involvements of these eIF3 components in a way beyond translation.

### dTak1 levels are regulated by eIF3f1

dTak1, a critical constituent of the IMD pathway, undergoes both the K63- and K48-linked ubiquitination modifications (Fernando et al, 2014). Pioneering studies have elucidated the functional involvement of the Dub dTrbd (*Drosophila* TRAF-binding domain-containing protein) in regulating the K63-linked ubiquitination of dTak1 (Fernando et al, 2014; Hua et al, 2022), However, the identity of the Dub orchestrating the K48-linked ubiquitination

regulation of dTak1 remains elusive. Herein, we furnish persuasive evidence indicating that eIF3f1 serves as a Dub that impedes the K48-linked ubiquitination of dTak1, thereby mediating its proteasome-dependent turnover. In mammals, TAK1 is known to be targeted by various Dubs, including CYLD, USP4, USP8, and USP18, for the dismantling of the K63-linked ubiquitin chains (Ahmed et al, 2011; Fan et al, 2011; Yang et al, 2015; Zhang et al, 2018). Additionally, USP19 targets both the K63- and the K27-linked ubiquitin chains on TAK1 (Lei et al, 2019). Nevertheless, a Dub that specifically targets the K48-linked ubiquitination of TAK1 has not yet been explicitly identified. Given our findings, it would be worthwhile to investigate whether the mammalian ortholog of *Drosophila* eIF3f1, eIF3F, is responsible for the K48-linked deubiquitination of TAK1. In addition, we would like to speculate two directions for future studies: (1) a time-course examination of the protein levels of eIF3f1/dTrbd/Posh during IMD signaling (or before/after bacterial infection); (2) investigation of the dynamics in the eIF3f1/dTrbd/Posh/dTak1 interaction network.

## Methods

### *Drosophila* strains

All flies were maintained on the standard *Drosophila* medium (6.65% cornmeal, 7.15% dextrose, 5% yeast, 0.66% agar, 2.2% nipagin, and 3.4 ml/l propionic acid) at 25 °C with a light/dark cycle of 12 h/12 h and 65% humidity. For genetic experiments employing the UAS/Gal4 system, crossings were first carried out at 18 °C. After the eclosion of the progenies, flies were collected and maintained at 29 °C for 5 to 7 d. The *eIF3f1 RNAi #1* and *eIF3f2 RNAi* strains were purchased from the Vienna *Drosophila* Resource Center (#101465 and #108169). The *eIF3f1 RNAi #2* and *dTak1 RNAi* strains were obtained from the Tsinghua RNAi Center (#1323 and #0756). The *UASp-GFP-eIF3f1^FL^, UASp-Flag-eIF3f1^JAMM^, UASp-Flag-eIF3f1^ΔJAMM^,* and *UASp-Flag-eIF3f2* transgenic flies were generated according to the P-element-based method (Bachmann and Knust, 2008). In brief, the coding sequence of *GFP-eIF3f1^FL^, Flag-eIF3f1^JAMM^, Flag-eIF3f11^ΔJAMM^,* or *Flag-eIF3f2* was inserted into the *UASp* empty vector. Each plasmid was then injected into the *w^1118^* embryos together with the *Δ2-3* plasmid, which expresses the enzyme for DNA recombination. Embryos were then laid on fruit juice plates and transferred into vials with normal medium at the larval stage. After eclosion, each fly was crossed separately, and the progenies were collected for transgene identification. The *c564-gal4* and *tub-gal80^ts^* strains were described previously (Zhu et al, 2023).

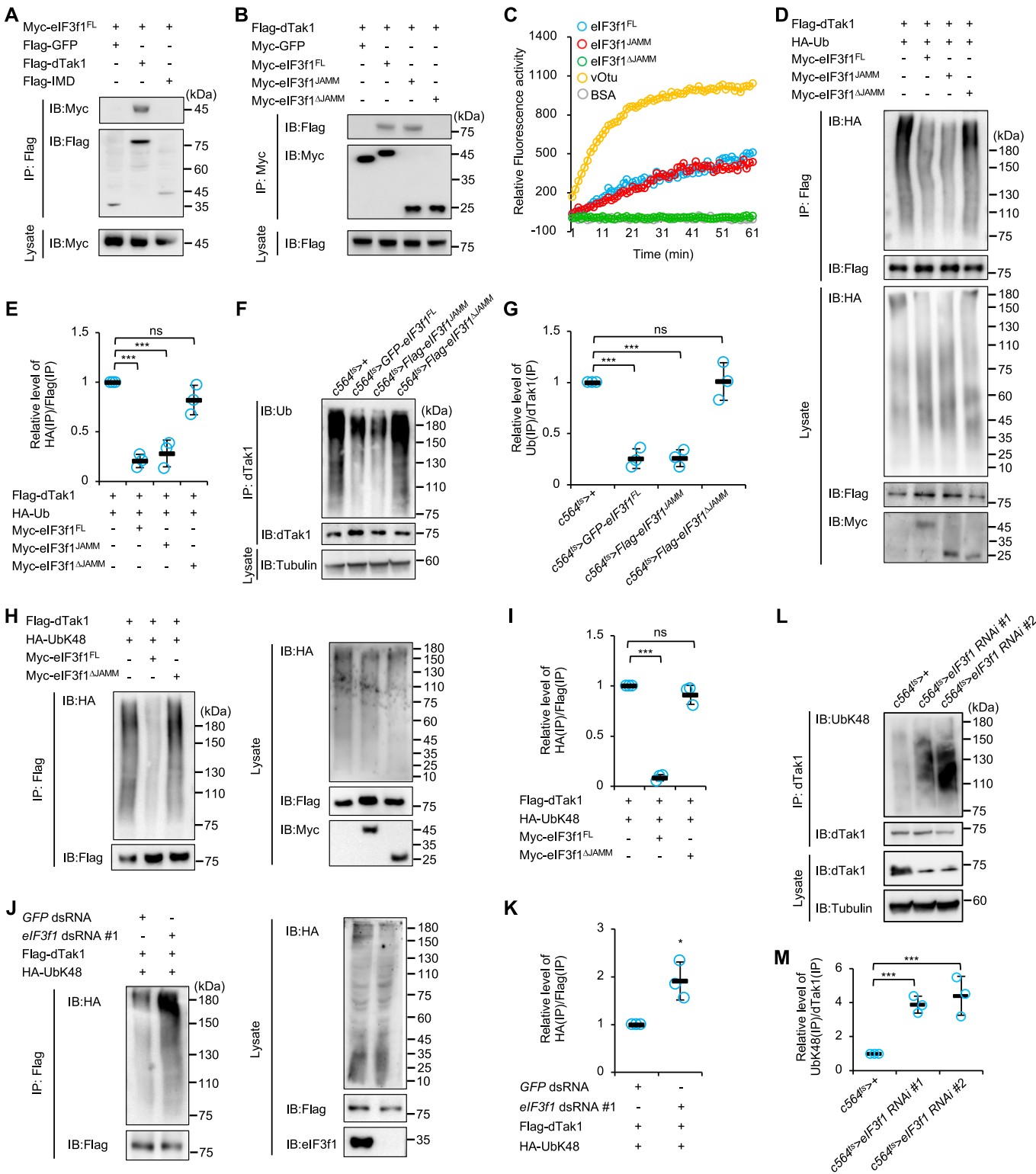

## Plasmids

The coding sequences of *eIF3f1^FL*, *eIF3f1^JAMM*, or *eIF3f11^ΔJAMM* were amplified from the cDNAs prepared from adult flies. Indicated DNA fragments were digested with NotI and XhoI, followed by ligation

reactions to insert them into the *pAc5.1* vector. The plasmids expressing Imd and dTak1 in S2 cells were established and described previously (Hua et al, 2022). The *pGL3-Att-Luc* and *pGL3-Drs-Luc* plasmids were synthesized by inserting the *attacin* promoter and *drosomycin* promoter into the *pGL3* vector, respectively (Ji et al, 2014).

**Figure 5. eIF3f1 associates with dTak1 and antagonizes the K48-linked ubiquitination of dTak1.**

(A) S2 cells were transfected with indicated plasmids for 48 h. Cells were lysed for co-immunoprecipitation assay. (B) Domain mapping analysis demonstrating that JAMM domain is both required and sufficient for eIF3f1 associating with dTak1. (C) In vitro deubiquitination assay examining the deubiquitinase enzymatical activities of indicated proteins. (D, E) S2 cells were transfected with indicated plasmids for 48 h. Cells were harvested for ubiquitination assays to examine the ubiquitination levels of dTak1. Densitometry analyses to quantify intensities of ubiquitinated dTak1 are shown in (E). (F, G) Fat body samples were collected from flies including $c564^{ts} > +$, $c564^{ts} > GFP\text{-}eIF3f1^{Fl}$, $c564^{ts} > Flag\text{-}eIF3f1^{JAMM}$, and $c564^{ts} > Flag\text{-}eIF3f1^{\Delta JAMM}$. Samples were lysed for ubiquitination assays. Quantification analyses of indicated proteins are shown in (G). (H, I) S2 cells were transfected with indicated plasmids for 48 h and harvested for ubiquitination assays. Quantification analyses of indicated proteins are shown in (I). (J, K) S2 cells were treated with various dsRNAs for 48 h and transfected with different plasmids for 48 h as indicated. Cells were lysed for ubiquitination assays. Quantification analyses of indicated proteins are shown in (I). (L, M) Fat body samples were collected from flies including $c564^{ts} > +$, $c564^{ts} > eIF3f1$ RNAi #1, and $c564^{ts} > eIF3f1$ RNAi #2. Samples were lysed for ubiquitination assays. Quantification analyses of indicated proteins are shown in M. Data Information: (E, G, I, K, M) Each dot represents one biological replicate. Data are shown as means ± SD. The nonparametric Kruskal–Wallis test was used for statistical analyses. (K) The Tukey's test was used for statistical analysis. *$P < 0.05$; ***$P < 0.001$; ns not significant, $P > 0.05$. Source data are available online for this figure.

## Antibodies

The following primary antibodies were utilized for Western blot assays: Rabbit anti-Myc (1:3000, Medical & Biological Laboratories, Cat#562); Mouse anti-β-Tubulin (1:3000, Cowin, Cat#CW0098M); Mouse anti-Flag (1:3000, Sigma-Aldrich, Cat#F1804); Rabbit anti-HA (1:3000, Medical & Biological Laboratories, Cat#561); Mouse anti-Puromycin (1:5000, Sigma-Aldrich, Cat#MABE343); Rabbit anti-dTak1 (1:1000, Abcam, Cat#239353) (Tsapras et al, 2022); Mouse anti-Ubiquitin (1:1000, Santa Cruz, Cat#sc-8017); Rabbit anti-UbK48 (1:1000, Sigma-Aldrich, Cat#SAB5701119); Mouse anti-eIF3f1 (1:2000), which was generated by immunizing mice with the purified N-terminal fragment of eIF3f1 (amino acids from 41 to 140).

The secondary antibodies utilized in this study include Goat anti-mouse IgG H&L (1:5000, Abcam, Cat#ab150078) and Goat anti-rabbit IgG H&L (1:5000, Abcam, Cat#ab6789).

## Bacterial infection, fly survival, and bacterial burden assay

The bacterial strains utilized for infection experiments include the Gram-negative bacteria *S. marcescens* (*Serratia marcescens*) and *Ecc15* (*Pectobacterium carotovorum carotovorum 15*) and the Gram-positive bacterium *E. faecalis* (*Enterococcus faecalis*). The *S. marcescens* and *E. faecalis* were obtained from the CGMCC (China General Microbiological Culture Collection Center), the numbers of which are 1.1215 and 1.2135, respectively. The *Ecc15* is a kind gift from Dr. Dominique Ferrandon's research group (Institut de Biologie Moléculaire et Cellulaire, France).

Bacterial infection was carried out according a protocol described previously (Cai et al, 2022). In brief, overnight bacterial cultures were harvested and diluted in sterile PBS buffer at a concentration of $OD_{600} = 1$. Male flies were collected and anesthetized on the fly pad with $CO_2$. The diluted bacteria (4.6 nl) or the same volume of PBS was injected into each fly with a nanoliter injector (Nanoject III, Drummond). After injection, flies were immediately transferred to fresh vials (30–40 flies per vial).

For the fly survival analysis, the numbers of death were counted every day. Flies that perished within 2 h (<5% of the total) after injection were not considered. Data were generated from three independent replicates and shown as means ± SD (standard errors).

For the bacterial burden assay, flies were collected, dipped in 75% EtOH, and volatilized with EtOH on the fly pad for several minutes. Flies were then homogenized separately in 100 µl sterile PBS, followed by serial dilutions. Finally, 100 µl of each diluent was incubated on an LB agar plate at 30 °C overnight. Data were collected from 21 independent replicates and pooled.

## S2 cell manipulation and dual-luciferase reporter assay

*Drosophila* S2 cells were grown in the insect medium (Gibco) supplemented with 10% FBS (fetal bovine serum, Hyclone) at 27 °C. Indicated plasmids were transfected into S2 cells utilizing the lipofectamine 2000 kit (Invitrogen). Thirty-six hours post transfection, cells were harvested and lysed in 50 µl of the passive lysis buffer (Promega). After centrifugation ($12{,}000 \times g$) at 4 °C for 10 min, 20 µl of the supernatant from each sample was added into a 96-well plate containing detection reagents (Promega). The Firefly and Renilla luciferase activities were monitored, and finally the ratio of Firefly to Renilla was calculated. Results were analyzed based on data from three independent biological replications.

For RNAi in S2 cells, dsRNAs were synthesized according to the manufacturer's protocol (Promega). Each dsRNA (3 µg) was added into S2 cells cultured in the FBS-free medium. One hour later, FBS was supplemented into these cells at a concentration of 10%. The primers for dsRNA syntheses are outlined in Table 1. The plasmids expressing Imd or the constitutively activated form of Toll (Toll$^{\Delta LRR}$) were utilized to induce the Att-Luc or Drs-Luc activity, respectively, in S2 cells.

We performed an unbiased genetic screening in S2 cells by using the Att-Luc reporter system. DsRNA treatments and transfections of plasmids were performed as described above. In this screening, we found that the Att-Luc level was reduced in the *eIF3f1* dsRNA-pretreated S2 cells, suggesting that *eIF3f1* potentially regulates the IMD signaling pathway. As there has not been any convincing evidence regarding the functional role of *eIF3f1* in modulating innate immunity, we decided to focus on investigating the potential immune function of *eIF3f1* and the underlying molecular mechanism.

## RT-qPCR assay

S2 cells were directly lysed in the Trizol reagent (Invitrogen). For the fat body tissues, samples were collected and homogenized in Trizol reagent with glass beads. Total RNA was extracted according to the standard chloroform/isopropanol method, followed by quality check and concentration calculation. The first-strand cDNA synthesis kit (Transgen) was used to reverse-transcribe RNA (1 µg) into cDNA. Quantitative PCR assays (in three technical repetitions) were carried out using the SYBR green supermix (Bio-Rad) in the CFX Opus platform (Bio-Rad). *RpL32* was used as the internal

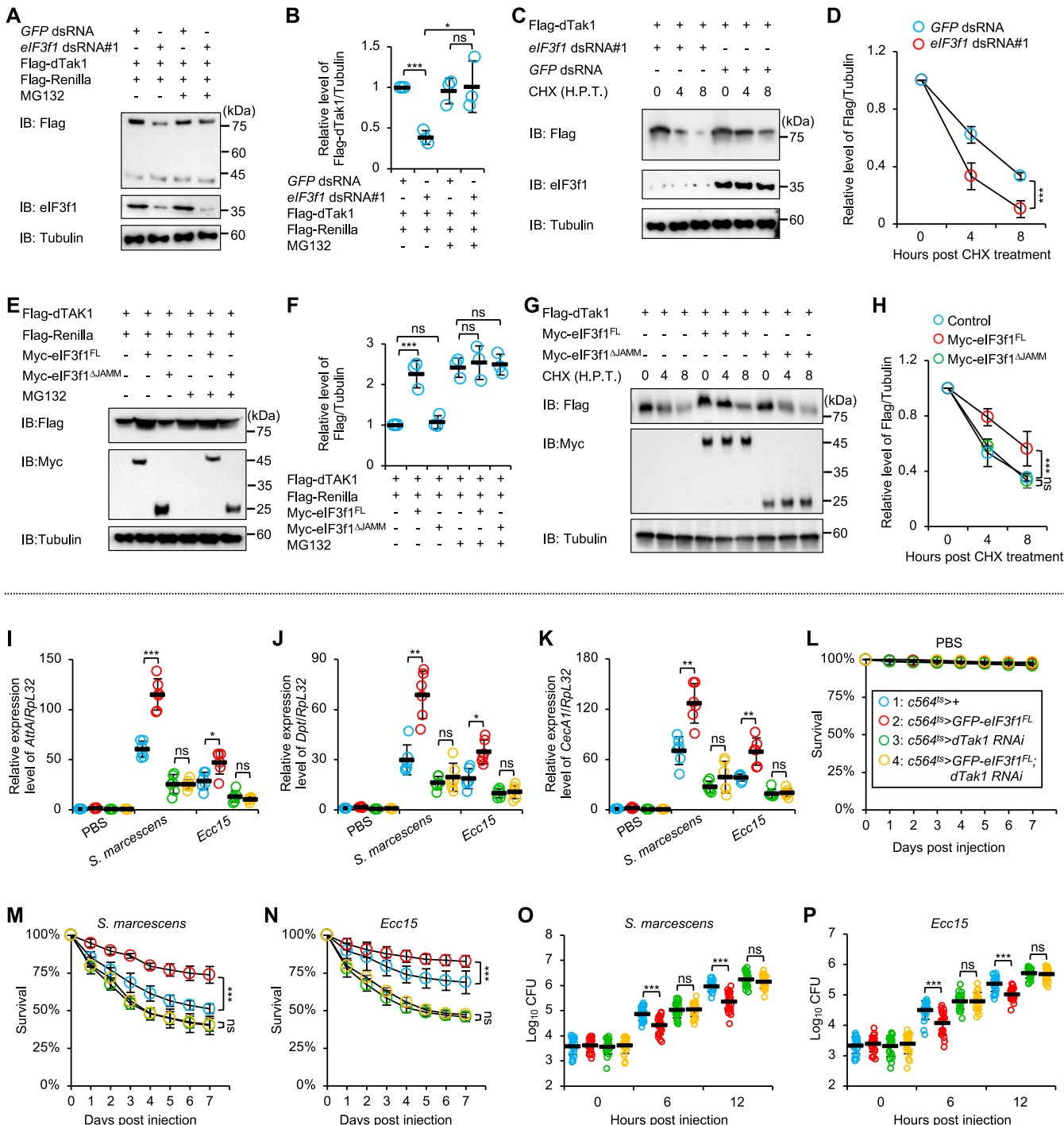

control. Results were analyzed based on data from six independent biological replicates. The primers used in RT-qPCR experiments are shown in Table 1.

## Western blot assay

S2 cells or fly samples were lysed in lysis buffer (150 mM NaCl, 50 mM Tris-HCl, pH = 7.5, 10% glycerol, 0.5% Triton X-100, and 1 mM phenylmethylsulfonyl fluoride). Samples were subjected to centrifugation (12,000× $g$) at 4 °C for 15 min. The supernatant was

subjected to the examination of protein concentration by Bradford assay. In total, 20 μg protein from each sample was loaded on an 8% to 15% gradient gel and separated by SDS-PAGE electrophoresis, followed by transferring to a PVDF membrane. The membrane was first blocked in PBST (0.1% Tween-20 in PBS) buffer with 5% bovine serum albumin for 30 min at room temperature, and then incubated with primary (at 4 °C overnight) and second (at room temperature for 1 h) antibodies as indicated. The blot was revealed by using the enhanced chemiluminescence substrate (Tiangen) in a Bio-Rad platform.

**Figure 6.    eIF3f1 targets dTak1 to mediate the fly antimicrobial immune defenses.**

(A, B) S2 cells were treated with various dsRNAs for 48 h and transfected with different plasmids for 36 h as indicated. Cells were then treated with MG132 (10 μM) or DMSO (control) for 6 h, and lysed for Western blot assays (A). Quantification analyses of indicated proteins are shown in (B). (C, D) S2 cells were treated with various dsRNAs for 48 h and transfected with dTak1 expressing plasmid for 36 h as indicated. Cells were then treated with CHX (50 μg/ml) or DMSO (control). At indicated time points (0, 4, and 8 h), cells were harvested for Western blot assays (C). Densitometry analyses to quantify dTak1 turnover are shown in (D). (E, F) S2 cells were transfected with different plasmids for 36 h as indicated. Cells were then treated with MG132 (10 μM) or DMSO (control) for 6 h, and lysed for Western blot assays (E). Quantification analyses of indicated proteins are shown in (F). (G, H) S2 cells were transfected with different expressing plasmid for 36 h as indicated. Cells were then treated with CHX (50 μg/ml) or DMSO (control). At indicated time points (0, 4, and 8), cells were harvested for Western blot assays (G). Densitometry analyses to quantify dTak1 turnover are shown in (H). (I–K) Flies including $c564^{ts} > +$, $c564^{ts} > GFP\text{-}eIF3f1^{FL}$, $c564^{ts} > dTak1\ RNAi$, and $c564^{ts} > GFP\text{-}eIF3f1^{FL};dTak1\ RNAi$ were injected with PBS, *S. marcescens*, or *Ecc15*. 6 h later, flies were harvested for RT-qPCR assays. (L–N) Flies including $c564^{ts} > +$, $c564^{ts} > GFP\text{-}eIF3f1^{FL}$, $c564^{ts} > dTak1\ RNAi$, and $c564^{ts} > GFP\text{-}eIF3f1^{FL};dTak1\ RNAi$ were injected with PBS (L), *S. marcescens* (M), or *Ecc15* (N), followed by survival analyses. The numbers of flies are as follows. (L) $c564^{ts} > +$: 91, 94, 93; $c564^{ts} > GFP\text{-}eIF3f1^{FL}$: 89, 90, 89; $c564^{ts} > dTak1\ RNAi$: 93, 88, 88; $c564^{ts} > GFP\text{-}eIF3f1^{FL};dTak1\ RNAi$: 94, 91, 91. (M), $c564^{ts} > +$: 86, 93, 86; $c564^{ts} > GFP\text{-}eIF3f1^{FL}$: 95, 85, 93; $c564^{ts} > dTak1\ RNAi$: 89, 83, 90; $c564^{ts} > GFP\text{-}eIF3f1^{FL};dTak1\ RNAi$: 92, 94, 87. (N) $c564^{ts} > +$: 93, 89, 86; $c564^{ts} > GFP\text{-}eIF3f1^{FL}$: 87, 92, 91; $c564^{ts} > dTak1\ RNAi$: 92, 91, 85; $c564^{ts} > GFP\text{-}eIF3f1^{FL};dTak1\ RNAi$: 91, 88, 87. (O, P) Flies including $c564^{ts} > +$, $c564^{ts} > GFP\text{-}eIF3f1^{FL}$, $c564^{ts} > dTak1\ RNAi$, and $c564^{ts} > GFP\text{-}eIF3f1^{FL};dTak1\ RNAi$ were injected with *S. marcescens* or *Ecc15*. At indicated time points, flies were collected for bacterial burden analyses. Data Information: (B, F, I, J, K, O, P) Each dot represents one biological replicate. Data are shown as means ± SD. The nonparametric Kruskal–Wallis test was used for statistical analyses. (D, H, M, N) The Log-Rank test was used for statistical analyses. *$P < 0.05$; **$P < 0.01$; ***$P < 0.001$; ns not significant, $P > 0.05$. Source data are available online for this figure.

## Examination of the translation efficiency in S2 cells

S2 cells were transfected with the eIF3f1 expressing plasmid or empty vector (control) as indicated. 36 h post transfection, cells were treated with cycloheximide (final concentration at 50 μg/ml) for 6 h. Cells were harvested and treated with 100 μl puromycin (10 μg/ml in PBS) solution for 10 min at 27 °C. Puromycin is an aminonucleoside antibiotic, which is produced by the bacterium *Streptomyces alboniger*. When cells were treated with proper amounts of puromycin, the nascent polypeptides can be incorporated by puromycin, which reflects directly the rate of global mRNA translation (Schmidt et al, 2009). Cells were then centrifuged (1000× g) at room temperature for 2 min, washed with complete medium (with 10% FBS) twice, and incubated in complete medium at 27 °C for 1 h. Finally, cells were lysed for Western blot assays utilizing anti-Puromycin antibodies. For the loading control, the protein level of Tubulin was monitored.

## Polysome fractionation by size exclusion chromatography

S2 cells were transfected with the eIF3f1 expressing plasmid or empty vector (control) for 48 h. Cells were then treated with cycloheximide (final concentration at 50 μg/ml) at 27 °C for 15 min, followed by centrifugation (1000× g) at room temperature for 2 min and wash treatment with ice-cold PBS containing 50 μg/ml cycloheximide. Samples were lysed in polysome buffer (10 mM Tris-HCl, pH = 7.5, 150 mM NaCl, 10 mM MgCl₂, 0.5% Nonidet P-40, 0.1 mg/ml cycloheximide, 2.5 mM dithiothreitol, 1 mM phenylmethylsulfonyl fluoride, and 500 u/ml RNase inhibitor) at 4 °C for 30 min. Cell lysates were centrifuged at 15,000× g for 10 min at 4 °C and filtered through a 0.45-μm Ultrafree-MC filter (Millipore). The protein and RNA concentrations in each sample were calculated by using the Pierce BCA Protein Assay kit (Thermo Fisher) and the Qubit™ RNA HS Assay kit (Thermo Fisher), respectively.

For the preparation of polysome fractionation, the BioBasic SEC 1000 Å column (Thermo Fisher) was equilibrated with 2 column volumes of filtered SEC buffer (10 mM Tris-HCl, pH = 7.5, 50 mM NaCl, 10 mM MgCl₂, 0.5% NP40, 2.5 mM dithiothreitol, 1 mM phenylmethylsulfonyl fluoride, and 500 u/ml RNase inhibitor).

Standards were injected into the columns to examine the condition. Later, cell lysates were injected into one column, followed by chromatogram measurements with UV absorbances at 215, 260, and 280 nm. The flow rate was 0.8 ml/min, and 16 × 300 fractions were collected. All column conditioning and separation were done at 4 °C.

To monitor the *dTak1* transcript levels in polysomes and monosomes, total RNAs were extracted using the Trizol and chloroform/isopropanol method as described above. RT-qPCR experiments were performed, and data were obtained from 6 independent biological replicates.

## Co-immunoprecipitation assay

S2 cells were transfected with indicated expressing plasmids. Forty-eight hours post transfection, cells were lysed in lysis buffer (150 mM NaCl, 50 mM Tris-HCl, pH = 7.5, 10% glycerol, 0.5% Triton X-100, and 1 mM phenylmethylsulfonyl fluoride) at 4 °C for 30 min. Samples were subjected to centrifugation (12,000× g) at 4 °C for 15 min. The supernatant was collected and incubated with Anti-Flag M2 affinity gel (Sigma) or anti-Myc affinity gel (Abmart) at 4 °C for 4 h. The immune complex was washed three times with wash buffer (500 mM NaCl, 50 mM Tris-HCl, pH = 7.5, 10% glycerol, and 0.5% Triton X-100) and subjected to the Western blot assay with indicated antibodies.

## In vitro deubiquitination assay

Purified proteins including vOtu (viral Otu, positive control), eIF3f1$^{FL}$, eIF3f1$^{JAMM}$, and eIF3f1$^{\Delta JAMM}$, or BSA (100 ng for each protein) was incubated with Ub-Rhodamine 110 (final concentration at 1 μM) in the reaction buffer (20 mM Tris-HCl, pH = 7.5, 200 mM NaCl, 5 mM MgCl₂, 2.5 mM dithiothreitol). Ub-Rhodamine 110 is an exquisitely sensitive Dub enzyme substrate for detecting ubiquitin C-terminal hydrolytic activity, because cleavage of the amide bond between the C-terminal glycine of ubiquitin and rhodamine by deubiquitinase leads to an increase in rhodamine fluorescence. The reaction mixture was added into a black 384-well plate and incubated at room temperature by using MD SpectraMax M5 Microplate Reader. Dynamic fluorescence was monitored with excitation and emission wavelengths set at 485/20

**Table 1. Primers used in this study.**

| Experiment | Name | Sequence |
|---|---|---|
| dsRNA synthesis | eIF3f1-#1-s | ACTATAGGGAGAATAGGTTAATGC |
| | eIF3f1-#1-as | ACTATAGGGAGATGCCTGTGGTAG |
| | eIF3f1-#2-s | ACTATAGGGAGAAGTGGTAAGGAC |
| | eIF3f1-#2-as | ACTATAGGGAGAAGAATTATCCGT |
| | eIF3f1-#3-s | ACTATAGGGAGAATAAGCGATAATTTA |
| | eIF3f1-#3-as | ACTATAGGGAGACTGATCATTAGTTTC |
| | GFP-s | ACTATAGGGAGAATGAGTAAAGGAGAA |
| | GFP-as | ACTATAGGGAGACATAACCTTCGGGCA |
| | Rel-s | ACTATAGGGAGAAGCAACGCCGAAACT |
| | Rel-as | ACTATAGGGAGAAGTACTACGACCTGG |
| | pll-s | ACTATAGGGAGACACAAGTACATACCG |
| | pll-as | ACTATAGGGAGAGCTGATGCTAAACCG |
| RT-qPCR | AttA-s | GGCCCATGCCAATTTATTCA |
| | AttA-as | AGCAAAGACCTTGGCATCCA |
| | Dpt-s | GCTGCGCAATCGCTTCTACT |
| | Dpt-as | TGGTGGAGTGGGCTTCATG |
| | CecA1-s | ACGCGTTGGTCAGCACACT |
| | CecA1-as | ACATTGGCGGCTTGTTGAG |
| | Drs-s | CGTGAGAACCTTTTCCAATATGATG |
| | Drs-as | TCCCAGGACCACCAGCAT |
| | Mtk-s | CAGTGCTGGCAGAGCCTCAT |
| | Mtk-as | ATAAATTGGACCCGGTCTTG |
| | RpL32-s | CACGATAGCATACAGGCCCAAGATCGG |
| | RpL32-as | GCCATTTGTGCGACAGCTTAG |
| | eIF3f1-s | GACGTTACCAACCACAGCTC |
| | eIF3f1-as | CAGACATAAGCACGCAGACC |
| | eIF3f2-s | CTGACAACGTAGTCGGGAGA |
| | eIF3f2-as | GCGAAATTTGTCCGAGTCCA |

Primers used in this study for dsRNA synthesis or RT-qPCR assays. "s" and "as" refer to "forward" and "reverse", respectively.

and 535/20 nm, respectively. Fluorescence intensity for each condition was averaged from triplicates and plotted as a function of time.

**Ubiquitination assay**

S2 cells were harvested, washed in ice-cold PBS twice, and lysed in 100 µl Ub lysis buffer (50 mM Tris-HCl, pH = 7.5, 500 mM NaCl, 0.5% Triton X-100, 10% glycerol, and 1% SDS). Samples were immediately boiled (98 °C) for 5 min. For dissected fat body tissues, samples were crushed in 100 µl Ub lysis buffer with glass beads. The liquid was carefully transferred to a new EP tube for boiling treatment (98 °C) for 5 min. In total, 900 µl of binding buffer (50 mM Tris-HCl, pH = 7.5, 500 mM NaCl, 0.5% Triton X-100, and 10% glycerol) was added into each sample to adjust SDS to a final concentration of 0.1%. Samples were then sonicated at 4 °C for 5 min and centrifugated (15,000× g) at 4 °C for 10 min. 10% of the supernatant was taken out and prepared as the lysate (input). The

rest (90%) of the supernatant was subjected to immunoprecipitation using anti-Flag M2 affinity gel agarose beads for 4 h. The immune complexes were washed three times using binding buffer for a total of 1 h, and subjected to Western blot assays to detect the ubiquitination patterns of indicated proteins.

**LC-MS/MS**

S2 cells were transfected with the eIF3f1 expressing plasmid or empty vector (control) for 48 h. Cells were harvested and lysed in lysis buffer (150 mM NaCl, 50 mM Tris-HCl, pH = 7.5, 10% glycerol, 0.5% Triton X-100, and 1 mM phenylmethylsulfonyl fluoride) at 4 °C for 30 min. Samples were precipitated with acetone at 4 °C overnight. The protein pellets were collected and digested with Trypsin and desalted using the PiecrceTMC-18 spin column (Thermo Fisher). Peptides were then subjected to LC-MS/MS and resulting MS/MS data were processed using Thermo Proteome Discovery (version 1.4.1.14) and searched against the UniProt-*Drosophila* database.

**Statistical analyses**

Statistical significances in Figs. 1A–H,M–Q, 2B–I,M–Q, 3D–H, 4F,H, 5E,G,I,M, 6B,F,I–K,O,P, EV1A–F, EV2A–F, EV3B–E, EV4B,D, and EV5A,C–F were determined by using the nonparametric Kruskal–Wallis tests. Statistical significances in Figs. 5K and EV4C were determined by using the Tukey's tests. Statistical significances in Figs. 1K,L, 2K,L, 3B,C, 6D,H,M,N, EV3A, and EV5B were calculated by using the Log-Rank tests. The $P$ value of less than 0.05 was considered statistically significant. *$P < 0.05$; **$P < 0.01$; ***$P < 0.001$.

## Data availability

The LC-MS/MS datasets produced in this study are available in the Mendeley Data: https://data.mendeley.com/preview/hcbx7rgmkf?a=0fd0b326-a51a-4d62-8436-65c068a79160.

## Peer review information

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

## Acknowledgements

We are grateful for the staff members at Omics-Laboratory of the Biotechnology Center of Anhui Agricultural University for providing technical support in mass data collection. We thank the Tsinghua RNAi Center and the Vienna Drosophila Resource Center for sharing fly strains. This work was supported by grants from the National Natural Science Foundation of China (32100702) and the Anhui Provincial Natural Science Foundation (2008085J14).

## Author contributions

**Yixuan Hu**: Resources; Data curation; Software; Formal analysis; Validation; Investigation; Visualization; Methodology; Writing—original draft; Writing—review and editing. **Fanrui Kong**: Resources; Data curation; Software; Formal analysis; Investigation; Methodology. **Huimin Guo**: Resources; Data curation; Software; Formal analysis; Investigation; Methodology. **Yongzhi Hua**: Resources; Investigation; Methodology. **Yangyang Zhu**: Resources; Investigation; Methodology. **Chuchu Zhang**: Resources; Investigation; Methodology. **Abdul Qadeer**: Resources; Software; Investigation; Methodology. **Yihua Xiao**: Resources; Investigation; Methodology. **Qingshuang Cai**: Conceptualization; Supervision; Validation; Visualization; Writing—original draft; Writing—review and editing. **Shanming Ji**: Conceptualization; Supervision; Funding acquisition; Writing—original draft; Project administration; Writing—review and editing.

## Disclosure and competing interests statement

The authors declare no competing interests.

# Expanded View Figures

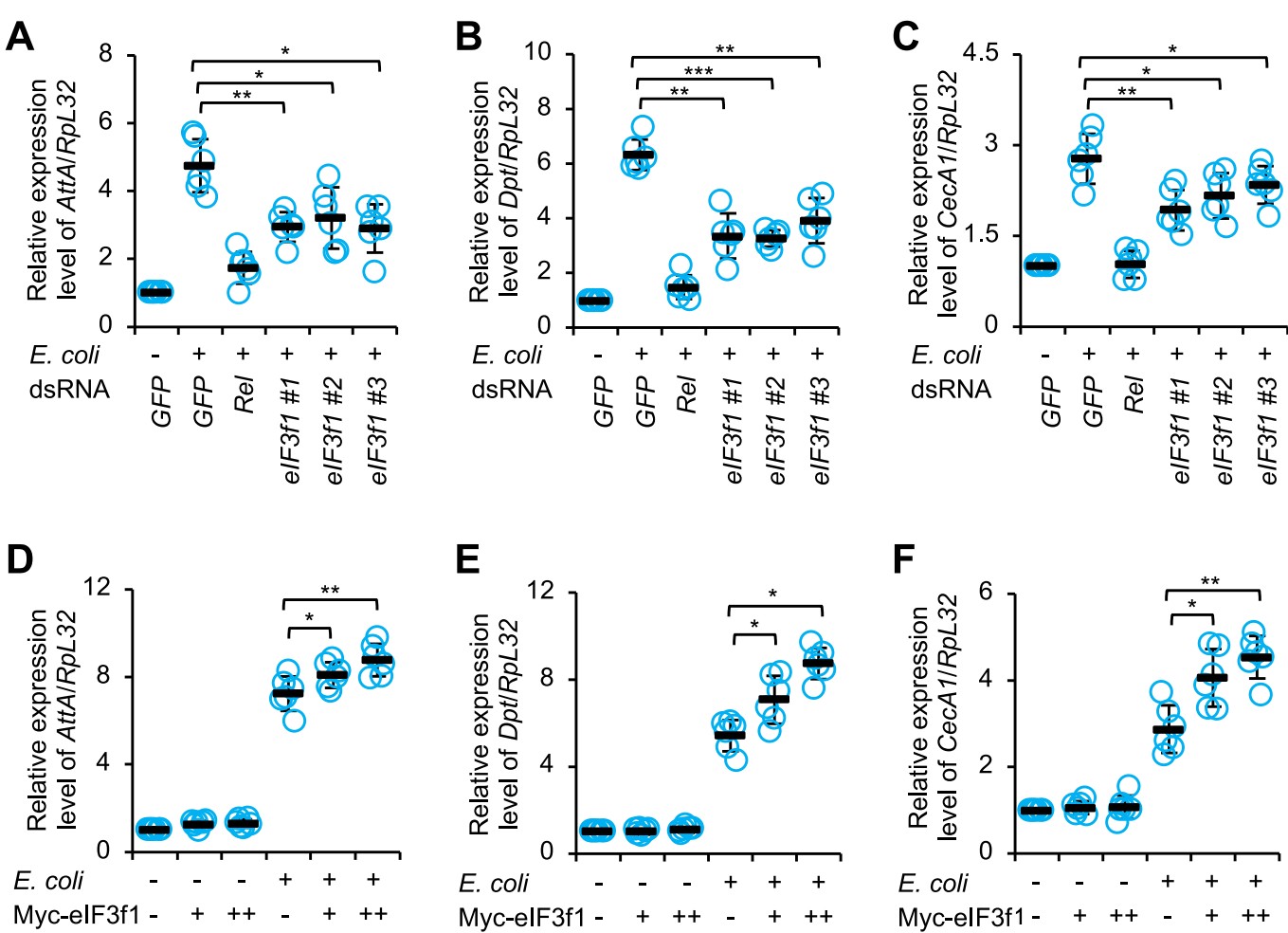

**Figure EV1.** *eIF3f1* modulates IMD signaling in *Drosophila*.

(**A–C**) S2 cells were treated with indicated dsRNAs for 48 h and then treated with heat-killed *E. coli* (MOI is around 5) for 6 h. Samples were subjected to RT-qPCR experiments to monitor the mRNA levels of *Att* (**A**), *Dpt* (**B**), or *CecA1* (**C**). (**D–F**) S2 cells were transfected with indicated expressing plasmids for 36 h, followed by treatment of heat-killed *E. coli* for 6 h. Samples were then subjected to RT-qPCR assays. Data Information: (**A–F**) Each dot represents one biological biological replicate and data are shown as mean ± SD. The nonparametric Kruskal–Wallis test was used for statistical analyses. *P < 0.05; **P < 0.01; ***P < 0.001.

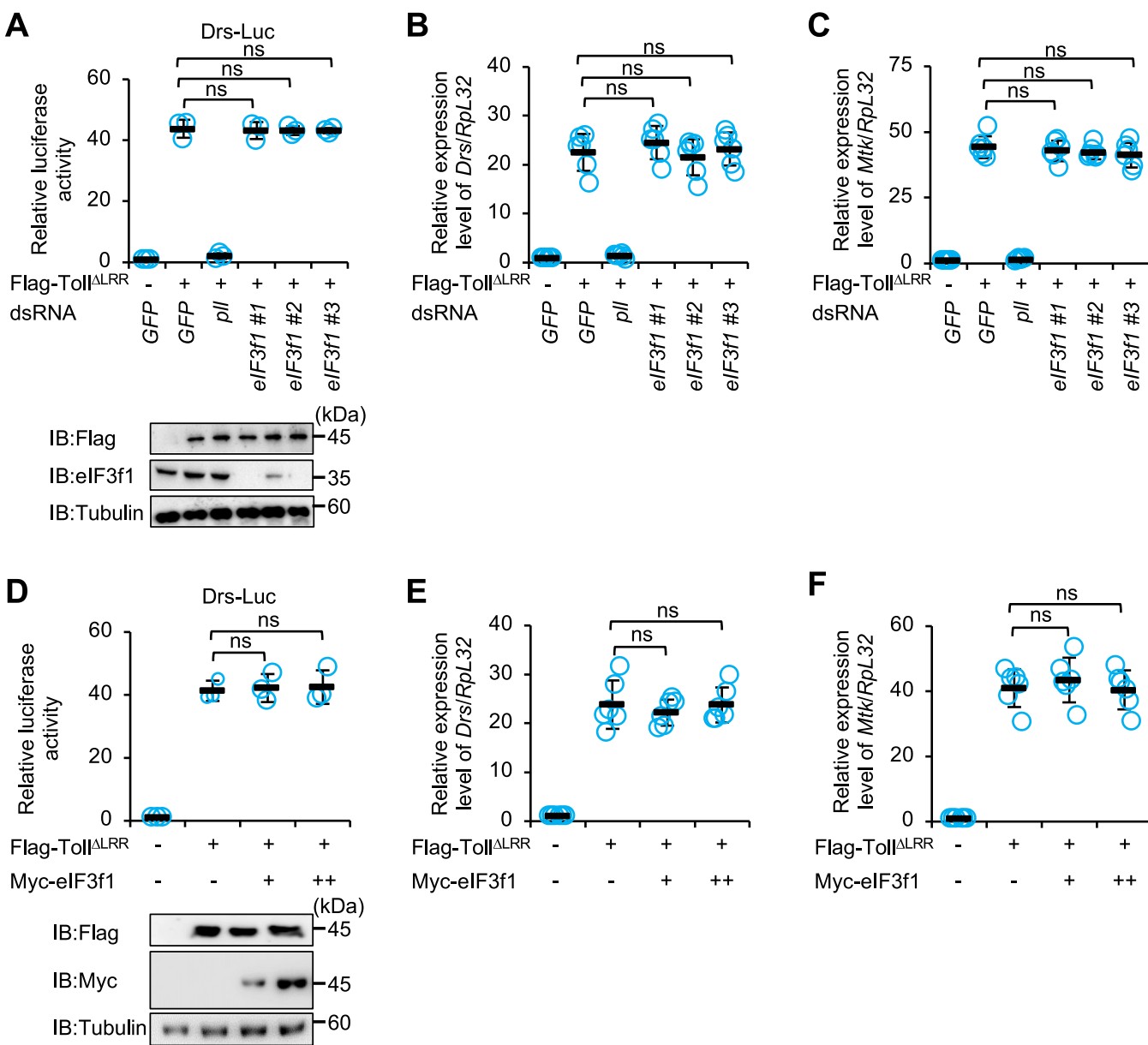

**Figure EV2. eIF3f1 does not affect Toll signaling in S2 cells.**

(A–C) S2 cells were treated with indicated dsRNAs for 48 h. Cells were then transfected with various combinations of expressing plasmids for 36 h, followed by dual-luciferase assays (A) or RT-qPCR experiments to monitor the mRNA levels of Drs (B) or Mtk (C). (D–F) S2 cells were transfected with indicated expressing plasmids. 36 h post transfection, cells were harvested for dual-luciferase (D) or RT-qPCR (E, F) assays. Data Information: (A–F) Each dot represents one biological replicate and data are shown as mean ± SD. The nonparametric Kruskal–Wallis test was used for statistical analyses. ns, not significant, P > 0.05.

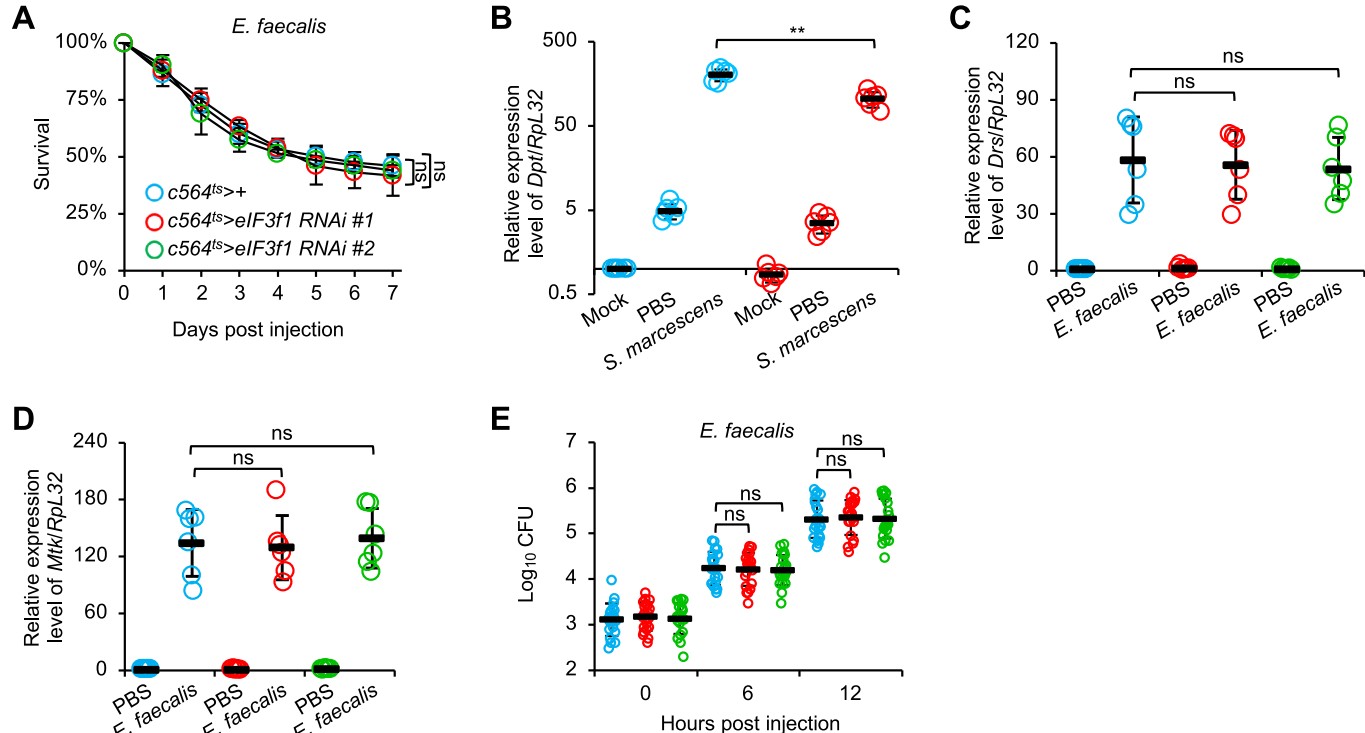

**Figure EV3. eIF3f1 is dispensable for the fly immune defense upon *E. faecalis* infection.**

(A–E) Adult flies including *c564^ts* > +, *c564^ts* > *eIF3f1 RNAi #1*, and *c564^ts* > *eIF3f1 RNAi #2* were infected with *E. faecalis* (**A, C–E**), *S. marcescens* (**B**), PBS (**B–D**), or without treatment (mock in **B**). Flies were subjected to survival analysis (**A**), or RT-qPCR assays (6 h post-infection, **B–D**), or bacterial burden assay (**E**). Data Information: (**A**) Data are shown as mean ± SD and the Log-Rank test was used for statistical analyses. The numbers of flies are as follows. *c564^ts* > +: 89, 85, 89; *c564^ts* > *eIF3f1 RNAi #1*: 84, 87, 85; *c564^ts* > *eIF3f1 RNAi #2*: 90, 92, 84. (**B–E**) Each dot represents one biological replicate and data are shown as mean ± SD. The nonparametric Kruskal–Wallis test was used for statistical analyses. **P < 0.01; ns, not significant, P > 0.05.

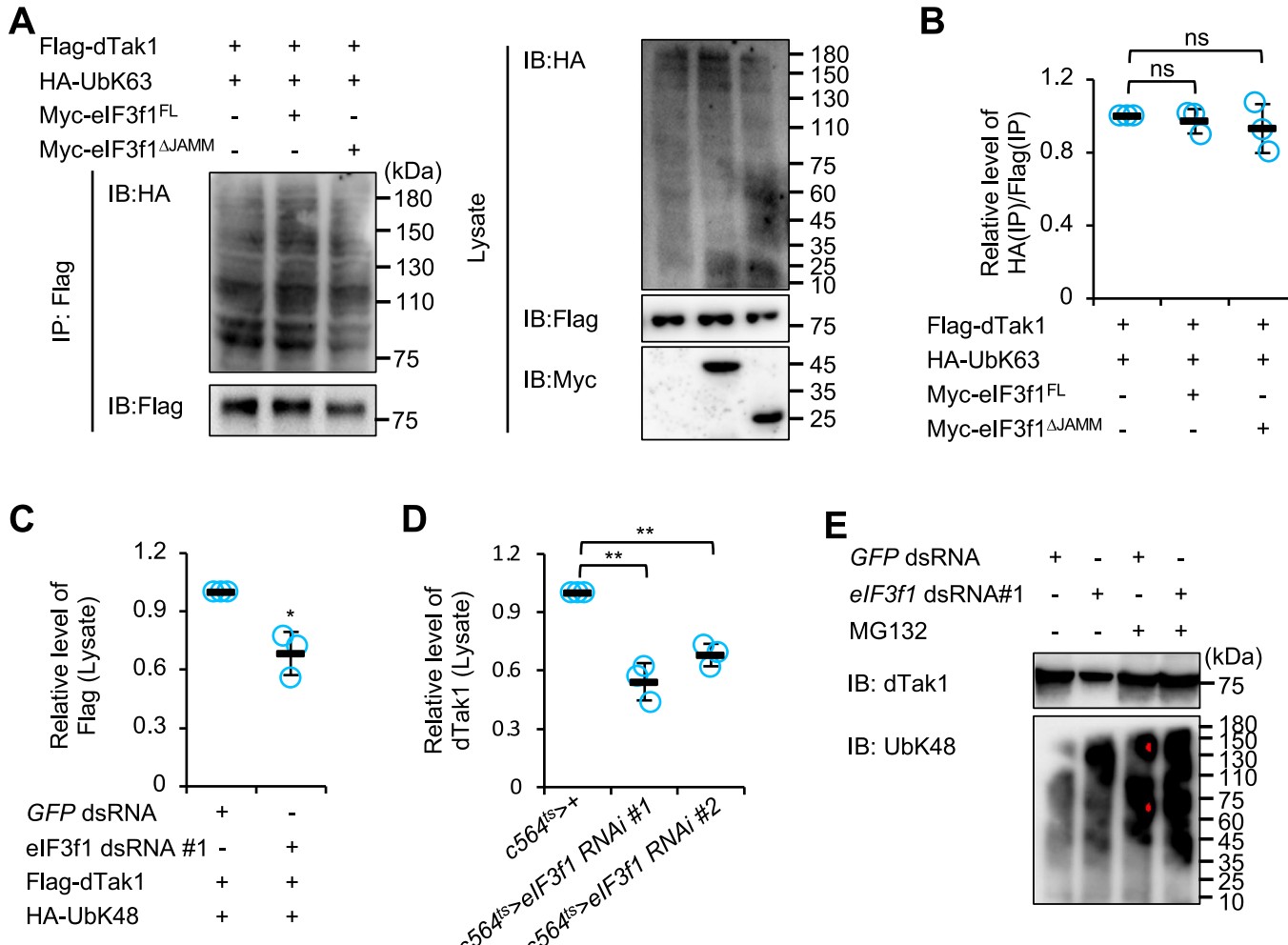

**Figure EV4.  eIF3f1 is not involved in regulating the K63-linked ubiquitination of dTak1.**

(A, B) S2 cells were transfected with indicated combinations of expressing plasmids. 48 h post transfection, cells were harvested for ubiquitination assay (A). Densitometry analysis to quantify the K63-linked ubiquitination levels of dTak1 is shown in (B). (C) Densitometry analysis to quantify Flag-dTak1 expression level in Fig. 5J. (D) Densitometry analysis to quantify dTak1 expression level in Fig. 5L. (E) S2 cells were treated with *GFP* or *eIF3f1* dsRNAs for 48 h, followed by MG132 treatment for 6 h as indicated. Cells were lysed for western blot assays using anti-dTak1 and anti-UbK48 antibodies. Data Information: (B, D) Each dot represents one biological replicate and data are shown as mean ± SD. The nonparametric Kruskal–Wallis test was used for statistical analyses. (C) The Tukey's test was used for statistical analysis. *P < 0.05; **P < 0.01; ns, not significant, P > 0.05.

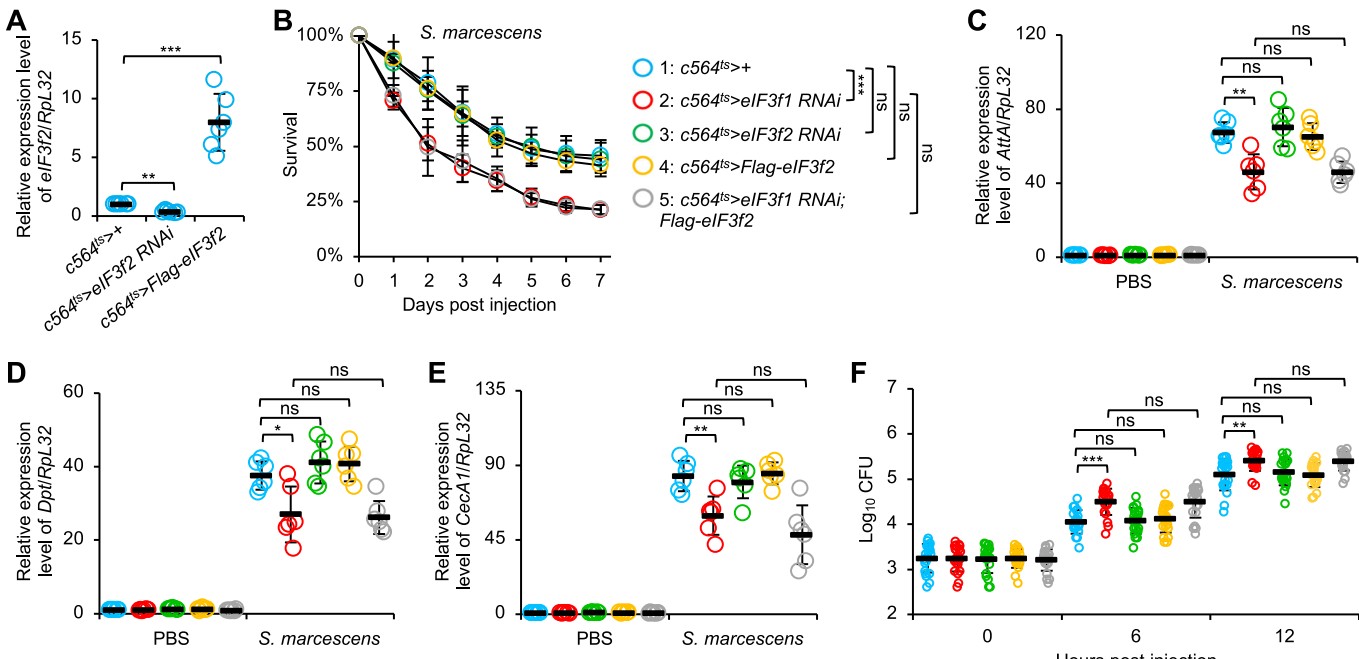

**Figure EV5. eIF3f2 is not involved in mediating the fly immune defense upon microbial infection.**

(A) RT-qPCR assays examining the knockdown or overexpression effect of *eIF3f2*. (B–F) Adult flies including *c564ts > +*, *c564ts > eIF3f1 RNAi*, *c564ts > eIF3f2 RNAi*, *c564ts > Flag-eIF3f2*, and *c564ts > eIF3f1 RNAi;Flag-eIF3f2* were infected with *S. marcescens* or PBS (control). Flies were subjected to survival analysis (B), or RT-qPCR assays (6 h post-infection, C–E), or bacterial burden assay (F). Data Information: (A, C–F) Each dot represents one biological replicate. Data are shown as mean ± SD. The nonparametric Kruskal–Wallis test was used for statistical analyses. (B) Data are shown as mean ± SD and the Log-Rank test was used for statistical analyses. The numbers of flies are as follows. *c564ts > +*: 89, 88, 94; *c564ts > eIF3f1 RNAi*: 91, 89, 84; *c564ts > eIF3f2 RNAi*: 96, 87, 89; *c564ts > Flag-eIF3f2*: 88, 89, 89; *c564ts > eIF3f1 RNAi;Flag-eIF3f2*: 89, 93, 94. *$P < 0.05$; **$P < 0.01$; ***$P < 0.001$; ns, not significant, $P > 0.05$.

