## [Peer Review File · EMBO Reports]

Drosophila eIF3f1 mediates the host immune defense through targeting dTak1

Yixuan Hu, Fanrui Kong, Huimin Guo, Yongzhi Hua, Yangyang Zhu, Chuchu Zhang, Abdul Qadeer, Yihua Xiao, Qingshuang Cai, and Shanming Ji

Corresponding author(s): Shanming Ji (jism@ahau.edu.cn)

Review Timeline:

Submission Date:	4th Aug 23
Editorial Decision:	1st Sep 23
Revision Received:	6th Nov 23
Editorial Decision:	7th Dec 23
Revision Received:	11th Dec 23
Editorial Decision:	9th Jan 24
Revision Received:	10th Jan 24
Accepted:	11th Jan 24

Editor: Achim Breiling

Transaction Report:

Dear Prof. Ji

Thank you for the submission of your research manuscript to our journal. We have now received the full set of referee reports that is copied below.

As you will see, the referees acknowledge that the findings are potentially interesting, but they also raise a number of concerns, all of which are important and need to be addressed.

Given these constructive comments, we would like to invite you to revise your manuscript with the understanding that the referee concerns (as detailed above and in their reports) must be fully addressed and their suggestions taken on board. Please address all referee concerns in a complete point-by-point response. Acceptance of the manuscript will depend on a positive outcome of a second round of review. It is EMBO Reports policy to allow a single round of revision only and acceptance or rejection of the manuscript will therefore depend on the completeness of your responses included in the next, final version of the manuscript.

We realize that it is difficult to revise to a specific deadline. In the interest of protecting the conceptual advance provided by the work, we recommend a revision within 3 months (December 1st). Please discuss the revision progress ahead of this time with the editor if you require more time to complete the revisions.

I am also happy to discuss the revision further via e-mail or a video call, if you wish.

IMPORTANT NOTE:

We perform an initial quality control of all revised manuscripts before re-review. Your manuscript will FAIL this control and the handling will be delayed IF the following APPLIES:

- 1) If a data availability section providing access to data deposited in public databases is missing. If you have not deposited any data, please add a sentence to the data availability section that explains that.
- 2) If your manuscript contains statistics and error bars based on $n=2$. Please use scatter blots in these cases. No statistics should be calculated if $n=2$.

- Please update the 'Conflict of interest' paragraph to our new 'Disclosure and competing interests statement'. For more information see

<https://www.embopress.org/page/journal/14693178/authorguide#conflictsofinterest>

- Regarding the Author Contributions, we now use CRediT to specify the contributions of each author in the journal submission system. CRediT replaces the author contribution section. Therefore, please remove the Author Contributions from the manuscript file and make sure that the author contributions in our online submission system are correct and up-to-date. The information you specified in the system will be automatically retrieved and typeset into the article. You can enter additional information in the free text box provided, if you wish.

- All Materials and Methods need to be described in the main text.

We would encourage you to use 'Structured Methods', our new Materials and Methods format. According to this format, the Materials and Methods section should include a Reagents and Tools Table (listing key reagents, experimental models, software and relevant equipment and including their sources and relevant identifiers) followed by a Methods and Protocols section in which we encourage the authors to describe their methods using a step-by-step protocol format with bullet points, to facilitate the adoption of the methodologies across labs. More information on how to adhere to this format as well as downloadable templates (.doc or .xls) for the Reagents and Tools Table can be found in our author guidelines:

< <https://www.embopress.org/page/journal/14693178/authorguide#manuscriptpreparation> >.

<<https://www.embopress.org/doi/10.15252/msb.20178071>>.

2) individual production quality figure files as .eps, .tif, .jpg (one file per figure).

Please download our Figure Preparation Guidelines (figure preparation pdf) from our Author Guidelines pages <https://www.embopress.org/page/journal/14693178/authorguide> for more info on how to prepare your figures.

4) a complete author checklist, which you can download from our author guidelines (<<https://www.embopress.org/page/journal/14693178/authorguide>>). Please insert information in the checklist that is also reflected in the manuscript. The completed author checklist will also be part of the RPF.

5) Please note that all corresponding authors are required to supply an ORCID ID for their name upon submission of a revised manuscript (<<https://orcid.org/>>). Please find instructions on how to link your ORCID ID to your account in our manuscript tracking system in our Author guidelines (<<https://www.embopress.org/page/journal/14693178/authorguide#authorshipguidelines>>)

6) We replaced Supplementary Information with Expanded View (EV) Figures and Tables that are collapsible/expandable online. A maximum of 5 EV Figures can be typeset. EV Figures should be cited as 'Figure EV1, Figure EV2" etc... in the text and their respective legends should be included in the main text after the legends of regular figures.

7) The Data Availability section should follow the Material and Methods section.

Please note that the Data Availability Section is restricted to new primary data that are part of this study. Please follow the template below and make sure that reviewer access is provided:

Data availability

Additional information on source data and instruction on how to label the files are available <<https://www.embopress.org/page/journal/14693178/authorguide#sourcedata>>.

10) Figure legends and data quantification:

- the name of the statistical test used to generate error bars and P values,

- the number (n) of independent experiments (please specify technical or biological replicates) underlying each data point,
- the nature of the bars and error bars (s.d., s.e.m.)

- If the data are obtained from n {less than or equal to} 5, show the individual data points in addition to the SD or SEM.
- If the data are obtained from n {less than or equal to} 2, use scatter blots showing the individual data points.

11) Our journal encourages inclusion of *data citations in the reference list* to directly cite datasets that were re-used and obtained from public databases. Data citations in the article text are distinct from normal bibliographical citations and should directly link to the database records from which the data can be accessed. In the main text, data citations are formatted as follows: "Data ref: Smith et al, 2001" or "Data ref: NCBI Sequence Read Archive PRJNA342805, 2017". In the Reference list, data citations must be labeled with "[DATASET]". A data reference must provide the database name, accession number/identifiers and a resolvable link to the landing page from which the data can be accessed at the end of the reference. Further instructions are available at <<https://www.embopress.org/page/journal/14693178/authorguide#referencesformat>>.

12) As part of the EMBO publication's Transparent Editorial Process, EMBO Reports publishes online a Review Process File to accompany accepted manuscripts. This File will be published in conjunction with your paper and will include the referee reports, your point-by-point response and all pertinent correspondence relating to the manuscript.

Yours sincerely,

Referee #1:

This manuscript present a curious finding suggesting that the translation factor, eIF3f1, thought to be a DUB, regulates the Drosophila immune response (the imd pathway) via a mechanism unrelated to translation regulation. instead the authors conduce the eIF3f1 directly de-K48 Ub TAK1, thereby stabilizing this critical MAP3K in the NF-κB pathway.

Major comments: the bulk of the data, while clear and compelling, relies on over-expression, especially all the molecular analysis showing TAK1 ubiquitination and stability, and the changes thereof with over-expressing eIF3f1 (such as Figure 5). Likewise, immune induction in cell lines were all driven with Imd or Toll over-expression (Figure 1 and more). Relying on over-expression often leads to artifacts. All key conclusions need to be demonstrated with bona fide immune challenges and manipulation of and assaying for endogenous proteins.

The results here should also be related to the literature. For example, the entire Imd pathway was analyzed for interactions via a tandem-tag, MS/MS type assays; was eIF3f1 found to be associated with TAK1 in this earlier study or any other study? On the flip side, do any other eIF3f components interact with TAK1 and/or modulate the Imd pathway? The authors argue for a direct interaction between TAK1 and eIF3f1, but other interpretations are possible, even likely, given their assay was performed with

over-expressed factors in a cognate (*Drosophila*) cell system.

With MG132 treatment, if the effect is really proteasome inhibition (and this inhibitor is not so specific to the proteasome), Ub intermediates should accumulate. Yet, they are not apparent in the data presented. Moreover, they should be K48Ub chains, if the conclusions are sound.

The genetic interactions shown in the 2nd half of Figure 6 do not support specifically the conclusion, that eIF3f1 regulates TAK1 stability/Ubiquitination, as similar results would like be observed with eIF3f1 over expression and RNAi targeting of any component of the Imd. pathway. The data is consistent with the model, but does not rule out other possibilities and again relies on over-expression of eIF3f1.

Minor comments:

Line 93: mentions Kenny RNAi as a control while the figure actually purports to show Relish RNAi. Note gene name for Relish includes a capital R, per flybase.

Line 160: how were the "rescue" experiment performed to avoid the effect of the RNAi on the transgenic version of the targeted gene?

Line 186: "We lent credit to the...." is a strange and unclear choice of language. please revise.

Line 200 and 348: the accuracy of the commercial dTAK1 antibody needs to be validated with a mutant or knockdown strain, or, at least, another paper with such validation cited. Commercial antibodies are notorious unreliable

Line 255: the effect is not so large to be considered "drastic", although it is significant.

Referee #2:

In this manuscript, authors have discovered the involvement of eIF3f1 (eukaryotic translation initiation factor 3 subunit f1) in the IMD pathway in *Drosophila*. They initially observed in the S2 cell system that knockdown and overexpression of eIF3f1 inhibit and enhance the activation of the IMD pathway, respectively. Subsequent in vitro and in vivo experiments demonstrated that this immunological function of eIF3f1 is mediated by the JAMM domain. Further mechanistic insights were obtained by proteomic analyses and biochemical studies, revealing that eIF3f1 interacts with dTAK1 and mediates the turnover of dTAK1 via K48-linked ubiquitination.

This study is grounded in a substantial amount of both genetic and biochemical data. The text is well written, the figures are well-organized and presented effectively. Undoubtedly, this study sets a high standard and contributes significantly to the field. Nevertheless, certain ambiguities and questions still remain. Once these points are clarified, the manuscript will definitely be suited for publication.

Major points:

1. It is still unclear how and why the authors initially chose to concentrate on eIF3f1 (L83-L87 and L99-L100): In references of 39 and 40, it is noted that not only eIF3f1 but also eIF3f2 and eIF3fh possess a JAMM domain. This raises at least two questions: 1) the rationale behind the authors' immediate focus on investigating the function of eIF3f1, and 2) the roles of other eIF3 subunits in the activation of the IMD pathway.

2. Regarding the aforementioned point: Based on the publication by Moretti et al in PLoS Biol (2010) (PMID21124883), it is noteworthy that mammalian eIF3h also contains a JAMM domain, and its intrinsic deubiquitinase activity regulates the Notch signaling pathway. While the report did not explore its involvement in innate immune signaling pathways, it seems that this manuscript might not be the first example to identify a role for a translation initiation factor at the level of post-translational regulation. Please clarify this point and if authors agree with this, tone down their claim.

3. Previous RNAi studies conducted in S2 cells have revealed that numerous genes implicated in translation exhibit impairments in cell viability. Even if the results are based on the normalization of either Actin-Renilla activity or Rpl32 expression, I am curious about the potential impact of eIF3f1 knockdown and overexpression on these normalization controls. If such an influence exists, it would be important to incorporate these observations within the manuscript and make necessary adjustments to certain interpretations.

Minor points:

1. Fig 1: Though authors elegantly showed their confirmation in vivo in Fig 2, The data of Fig 1 are all based in the context of IMD overexpression. How about the phenotype on the physiological heat-killed *E. coli* stimulated condition? Plus, in Fig 2E-H, there is no data point of simple eIF3f1 overexpression.

2. Fig 4A and L198-L199: What are the two bands (~65kDa and ~40 kDa) corresponding to? What is the experimental system of using puromycin-involved translation examination assay? Though the detail may be described in reference 44, it is difficult for

general reader like me to understand the system.

3. Fig 5J: I do not see a significant reduction of dTAK1 like the case of Fig 6A. Clarify the results in a quantitative manner and explain the possible reasons.
4. Fig 6E & F: I do not see the bands and quantifications of control Flag-Renilla.
5. While this manuscript centers on eIF3f1, there appears to be a lack of comprehensive information concerning the broader role of eIF3s in translation. How does this complex participate in translation? In what ways does this function diverge from other eIFs? What are the sequence similarities and domain compositions when compared to mammalian eIF3s? etc. Incorporating such details, even if the findings extend beyond of translation, it would be valuable and should be included either within the introduction or discussion section.
6. After having this study, how do author expect the relationship between eIF3f1, dTrabid and Posh? If authors have any data and/or speculations, it will be valuable to describe in the discussion section.
7. L104 and L407: But, the data is not Kenny but Relish?
8. Fig4C: Why is there no comment regarding carbohydrate derivative biosynthetic & metabolic processes?
9. Fig 4D: Not only top 10 candidates, but also a full list of candidates in the supplemental table will be very much useful to the community.
10. Fig 5C: What could be the reason why eIF3f1 has a lower deubiquitination activity as compared to a positive control vOtu? What could be the determinants of deubiquitination activity? Is this because of the structural difference or sequence conservation?
11. L498: μ not u ?
12. In materials and methods part, no section for the plasmids?

Referee #3:

In this article, the authors analyze the function in the *Drosophila* host defense against bacterial infection of a putative translation initiation factor, eIF3f1, that contains a JAMM domain thought to mediate deubiquitylation reactions. They first showed both in cultured cells and in vivo by RNAi silencing that the loss-of-function phenotype is a reduced induction of antimicrobial peptide (AMP) genes that correlated with a mildly increased sensitivity to a couple of Gram-negative bacterial infections, consistent with the Immune-deficiency (IMD) pathway being affected, and not the Toll pathway. Next, by relying on overexpression experiments both in cultured cells and in vivo, they showed that the N-terminal JAMM domain is required to mediate an increased IMD-dependent immune response, again the effects being relatively mild. Of note, the overexpression of eIF3f1 rescues the loss-of-function phenotype. A second part of the manuscript deals with the effect on translation and protein stability of eIF3f1 overexpression. This overexpression appeared to increase translation in cultured cells and one of the proteins the level of which is most strongly expressed is the TAK1 kinase, which is a component of the IMD pathway. Interestingly, the increased steady-state levels of dTAK1 are apparently not due to increased protein translation but to decreased proteasome-mediated degradation due to the deubiquitylation of K48-ubiquitin chains on TAK1 mediated by the JAMM domain. Finally, the effects of eIF3f1 overexpression could be alleviated by silencing TAK1, which suggests that genetically TAK1 functions downstream of eIF3f1.

While this study is generally well-conducted, a number of outstanding issues remain.

An important claim is that it is the first time that a non-canonical function in immunity is described for a eukaryotic translation initiation factor. While the evidence appears to indeed support a role for this factor in the IMD pathway, although some additional checks are needed (see below), the evidence that eIF3f1 is indeed a translation initiation factor is limited to the overexpression experiment shown in Fig. 4A and implicitly to its homology of sequence to mammalian eIF3f. The authors do not at all mention that an apparent specificity of *Drosophila melanogaster* (and potentially other Dipteran or insect/arthropod species, this reviewer did not take the time to check) is that the genome encodes two paralogs of this factor. eIF3f2 displays 30% identity and some 60% similarity to eIF3f1, which spans rather homogeneously the whole protein. Can the authors exclude that following the duplication of the locus there has been an evolutionary divergence of function, eIF3f2 being the actual translation initiation factor for instance and eIF3f1 taking on an immune function, possibly among others? The authors should check whether the eIF3f2 factor also contains a JAMM domain and let the reader know whether the mammalian factor also contains this domain. The authors should show that eIF3f1, or eIF3f2, is a bona fide translation initiation factor using a loss-of-function approach as overexpression may be misleading. What happens to TAK1 protein levels when these factors are silenced? A related issue to the existence of the paralog is that it is imperative that the authors experimentally demonstrate that their silencing constructs specifically affect eIF3f1 and not eIF3f2 expression, which may be the case according to VDRC since no

computed off-target effect is reported.

If the functions of the two paralogs are indeed distinct, the overexpression of eIF3f2 may provide an interesting control. One point that would enhance the appeal of the second part of this study would be the description of what happens when immune signaling is triggered, especially with respect to TAK1 levels and K48-ubiquitinylation in wild-type conditions or when eIF3f1 is either overexpressed or silenced. It would also be worth directly monitoring AMP levels by mass-spectrometry on collected hemolymph (Uttenweiler-Joseph et al, PNAS, 1998) in the different conditions after a mixed Gram-positive and Gram-negative challenge so as to induce both the IMD and the Toll pathway (Drosomycin and Bomanins will serve as internal nonaffected controls, thus bypassing the fact that MALDI-TOF is not especially well-suited for quantitation).

Minor points:

- The title should contain the term *Drosophila*.
- Refs 18 and 19 are not the most relevant when referring to PGRP-LC and-LE. Besides citing the articles that documented their function, the authors ought to cite the structural work that described the direct binding of peptidoglycan fragments to these receptors. The alternative is to cite a review article.
- Line 77 and others: "silencing" and not "silence"
- Line 94: Kenny is NOT a kinase, just a key adaptor, which binds to K63 polyubiquitin chains...
- The authors should mention that the c564 driver is not solely expressed in the fat body
- The authors should check whether the melanization and the cellular immune response are affected when eIF3f1 is silenced in flies.
- In all figures documenting the expression of AMP genes, it is disturbing that the PBS injection controls do not appear to induce their expression, which has been largely documented. The authors should include nonchallenged flies in at least one experiment.
- Line 192: "dramatically"; line 197: "considerable": please, avoid using such hyperbolic terms especially when they do not mirror the data
- Line 211: "indicate" instead of "intimate"
- FIG. 4D: can the authors exclude that all of their other hits reflect the deubiquitinylation properties of eIF3f1?
- The effects of TAK1 silencing appear to be rather mild as compared to those of a null mutant. Can the authors please assess the effectiveness of their RNAi line and mention the limitation of RNAi, the risk of generating hypomorphic phenotypes?
- Statistics: the type of statistics used should be described for each panel (ANOVA or Mann-Whitney, and why they use it). Actually, except for Fig. 4F and 5K, the authors perform multiple comparisons: they should use the Kruskal-Wallis test for non-parametric tests and also perform post-hoc tests for all multiple comparison tests).

Responses to comments from the editor and referees

Editor:

Thank you for the submission of your research manuscript to our journal. We have now received the full set of referee reports that is copied below. As you will see, the referees acknowledge that the findings are potentially interesting, but they also raise a number of concerns, all of which are important and need to be addressed. Given these constructive comments, we would like to invite you to revise your manuscript with the understanding that the referee concerns (as detailed above and in their reports) must be fully addressed and their suggestions taken on board. Please address all referee concerns in a complete point-by-point response. Acceptance of the manuscript will depend on a positive outcome of a second round of review. It is EMBO Reports policy to allow a single round of revision only and acceptance or rejection of the manuscript will therefore depend on the completeness of your responses included in the next, final version of the manuscript.

We realize that it is difficult to revise to a specific deadline. In the interest of protecting the conceptual advance provided by the work, we recommend a revision within 3 months (December 1st). Please discuss the revision progress ahead of this time with the editor if you require more time to complete the revisions.

I am also happy to discuss the revision further via e-mail or a video call, if you wish.

Response: We deeply appreciate that both the editor and referees spent their precious time thoroughly evaluating our manuscript and positively commented on our work. We are grateful for the constructive concerns which help us to further improve the quality of our paper. The detailed point-by-point responses to referees' comments are outlined below.

Referee #1:

This manuscript presents a curious finding suggesting that the translation

factor, eIF3f1, thought to be a DUB, regulates the *Drosophila* immune response (the imd pathway) via a mechanism unrelated to translation regulation. Instead, the authors conclude that eIF3f1 directly de-K48 Ub TAK1, thereby stabilizing this critical MAP3K in the NF- κ B pathway.

Major comments:

1. The bulk of the data, while clear and compelling, relies on over-expression, especially all the molecular analysis showing TAK1 ubiquitination and stability, and the changes thereof with over-expressing eIF3f1 (such as Figure 5). Likewise, immune induction in cell lines were all driven with Imd or Toll over-expression (Figure 1 and more). Relying on over-expression often leads to artifacts. All key conclusions need to be demonstrated with bona fide immune challenges and manipulation of and assaying for endogenous proteins.

Response: We thank the referee for the positive feedback. To illustrate the functional involvement of *eIF3f1* in mediating the IMD signaling pathway, we first examined the phenotype of *eIF3f1* in S2 cells with Imd over-expression (Figs 1A to H and 2A to I). To address the referee's concerns (also raised by referee #2), we challenged S2 cells with heat-killed *E. coli* and obtained consistent results (Fig EV1A to F in the revision). We also performed a series of experiments on *eIF3f1* RNAi or over-expression flies with bona fide immune challenges, and have shown these data in Figs 1J to Q and 2J to Q.

Regarding the molecular mechanism by which eIF3f1 mediates *Drosophila* IMD innate immune response, we have shown that eIF3f1 specifically targets dTak1 at the post-translational but not translational level. For this, we have provided *in vivo* evidence demonstrating that ectopic eIF3f1 prevents endogenous dTak1 ubiquitination and degradation (Fig 5F and G). In the revised manuscript, we have included results displaying that silencing *eIF3f1* leads to enhanced ubiquitination and decreased stability of endogenous dTak1 in the *Drosophila* fat body (Figs 5L, M and EV4D).

2. The results here should also be related to the literature. For example, the entire Imd pathway was analyzed for interactions via a tandem-tag, MS/MS type assays; was eIF3f1 found to be associated with TAK1 in this earlier study or any other study? On the flip side, do any other eIF3f components interact with TAK1 and/or modulate the Imd pathway? The authors argue for a direct interaction between TAK1 and eIF3f1, but other interpretations are possible, even likely, given their assay was performed with over-expressed factors in a cognate (*Drosophila*) cell system.

Response: We thank the referee for these excellent comments, which were also raised by referees #2 and #3. Several previous studies (in different species) have utilized TAK1 as a bait to investigate the TAK1 interactome (Fukuyama et al, Proc Natl Acad Sci USA, 2013, PMID: 23749869; Kajino et al, J Biol Chem, 2007, PMID: 17276978; Sokolova et al, Oncotarget, 2018, PMID: 29581850). However, no data have so far supported the physical interaction between dTak1 and eIF3f1, before our investigation. We deeply agree with the referee that our current findings cannot support a direct physical interaction of dTak1/eIF3f1. Therefore, we have toned down our conclusions in the revised version of our manuscript.

To address whether the other eIF3f component (eIF3f2) is involved in modulating the fly IMD innate immune defense, we subjected the *c564^{ts}>eIF3f2 RNAi* flies for bacterial challenging. As demonstrated in Fig EV5A to E in the revision, the down-regulation of *eIF3f2* hardly affected the AMP expressions and fly survival upon microbial infection. In addition, we generated the *eIF3f2* over-expression transgenic fly and found that ectopic *eIF3f2* did not impact on the host antimicrobial immune defenses (Fig EV5A to E). We have included these data in the revised version of our manuscript.

3. With MG132 treatment, if the effect is really proteasome inhibition (and this inhibitor is not so specific to the proteasome), Ub intermediates should

accumulate. Yet, they are not apparent in the data presented. Moreover, they should be K48Ub chains, if the conclusions are sound.

Response: In agreement with the referee's idea, our results have clearly demonstrated that eIF3f1 mediates the K48-linked but not the K63-linked ubiquitination of dTak1 (Figs 5H to K and EV4A and B).

4. The genetic interactions shown in the 2nd half of Figure 6 do not support specifically the conclusion, that eIF3f1 regulates TAK1 stability/Ubiquitination, as similar results would like be observed with eIF3f1 over expression and RNAi targeting of any component of the Imd pathway. The data is consistent with the model, but does not rule out other possibilities and again relies on over-expression of eIF3f1.

Response: We thank the referee for pointing out this issue. We have toned down our conclusions in the revised manuscript.

Minor comments:

1. Line 93: mentions Kenny RNAi as a control while the figure actually purports to show Relish RNAi. Note gene name for Relish includes a capital R, per flybase.

Response: We feel sorry for this mistake and have modified the gene name accordingly.

2. Line 160: how were the "rescue" experiment performed to avoid the effect of the RNAi on the transgenic version of the targeted gene?

Response: We are aware that this is not a perfect way and it would be better to use the *eIF3f1* loss-of-function mutants, which are unfortunately not viable. Instead, we utilized the *eIF3f1* RNAi strain for the rescue experiments, a manner that worked out in a series of previous studies (e.g., Cui and DiMario, Mol Biol Cell, 2007, PMID: 17392509; Lindstrom et al, BMC Genet, 2017, PMID: 28578657; Qiao et al, Nat Commun, 2018, PMID: 30297884; Zhou et al,

Sci Adv, 2019, PMID: 31555733). We reason that this worked out because the *eIF3f1 RNAi* cannot efficiently block all the expressions of ectopic *eIF3f1*.

3. Line 186: "We lent credit to the..." is a strange and unclear choice of language. please revise.

Response: We have changed this description into "We performed the puromycin-involved translation examination assay".

4. Line 200 and 348: the accuracy of the commercial dTAK1 antibody needs to be validated with a mutant or knockdown strain, or, at least, another paper with such validation cited. Commercial antibodies are notorious unreliable.

Response: According to the referee's suggestion, we have included the reference (Tsapras et al, Cell Rep, 2022, PMID: 35081354) in the revised manuscript.

5. Line 255: the effect is not so large to be considered "drastic", although it is significant.

Response: We have deleted these descriptive words as suggested.

Referee #2:

In this manuscript, authors have discovered the involvement of eIF3f1 (eukaryotic translation initiation factor 3 subunit f1) in the IMD pathway in *Drosophila*. They initially observed in the S2 cell system that knockdown and overexpression of eIF3f1 inhibit and enhance the activation of the IMD pathway, respectively. Subsequent in vitro and in vivo experiments demonstrated that this immunological function of eIF3f1 is mediated by the JAMM domain. Further mechanistic insights were obtained by proteomic analyses and biochemical studies, revealing that eIF3f1 interacts with dTAK1 and mediates the turnover of dTAK1 via K48-linked ubiquitination. This study is grounded in a substantial amount of both genetic and biochemical data. The

text is well written, the figures are well-organized and presented effectively. Undoubtedly, this study sets a high standard and contributes significantly to the field. Nevertheless, certain ambiguities and questions still remain. Once these points are clarified, the manuscript will definitely be suited for publication.

Major points:

1. It is still unclear how and why the authors initially chose to concentrate on eIF3f1 (L83-L87 and L99-L100): In references of 39 and 40, it is noted that not only eIF3f1 but also eIF3f2 and eIF3fh possess a JAMM domain. This raises at least two questions: 1) the rationale behind the authors' immediate focus on investigating the function of eIF3f1, and 2) the roles of other eIF3 subunits in the activation of the IMD pathway.

Response: We initiated this work based on a genetic screening in S2 cells utilizing the Att-Luc reporter system. In this screening, we identified *eIF3f1* as a potential modulator in the IMD signaling pathway. We further conducted both *in vitro* and *in vivo* investigations to address the functional assessment of *eIF3f1* in the fly antimicrobial immune defense and the underlying molecular mechanism.

Regarding the potential contribution of *eIF3f2* to regulating the innate immune defense in *Drosophila* (also raised by referee #1), we have performed additional experiments showing that neither the down-regulation nor the over-expression of *eIF3f2* affects the fly innate immune reactions upon bacterial infections (Fig EV5A to E in the revision). To our best knowledge, it has remained largely unknown whether other eIF3 subunits are involved in modulating the IMD signaling pathway. We definitely agree with the referee and would like to explore the potential functional roles of other eIF3 components in *Drosophila* innate immunity in our future projects.

2. Regarding the aforementioned point: Based on the publication by Moretti et

al in PLoS Biol (2010) (PMID21124883), it is noteworthy that mammalian eIF3h also contains a JAMM domain, and its intrinsic deubiquitinase activity regulates the Notch signaling pathway. While the report did not explore its involvement in innate immune signaling pathways, it seems that this manuscript might not be the first example to identify a role for a translation initiation factor at the level of post-translational regulation. Please clarify this point and if authors agree with this, tone down their claim.

Response: We are grateful for the referee pointing out these issues. In the revised manuscript, we have cited this paper and toned down our claims.

3. Previous RNAi studies conducted in S2 cells have revealed that numerous genes implicated in translation exhibit impairments in cell viability. Even if the results are based on the normalization of either Actin-Renilla activity or RpL32 expression, I am curious about the potential impact of eIF3f1 knockdown and overexpression on these normalization controls. If such an influence exists, it would be important to incorporate these observations within the manuscript and make necessary adjustments to certain interpretations.

Response: In fact, we didn't observe marked alterations regarding Renilla activities and *RpL32* expression levels, when we performed dual-luciferase and RT-qPCR assays under the conditions of *eIF3f1* RNAi or over-expression.

Minor points:

1. Fig 1: Though authors elegantly showed their confirmation in vivo in Fig 2, The data of Fig 1 are all based in the context of IMD overexpression. How about the phenotype on the physiological heat-killed *E.coli* stimulated condition? Plus, in Fig 2E-H, there is no data point of simple eIF3f1 overexpression.

Response: To address this concern (also raised by referee #1), we have included in the revised manuscript the data using *E. coli* stimulation (Fig EV1A to F). In addition, we performed experiments examining the relative luciferase activities with only *eIF3f1* over-expression. These results have been shown in

Fig EV1D to F in the revision.

2. Fig 4A and L198-L199: What are the two bands (~65kDa and ~40 kDa) corresponding to? What is the experimental system of using puromycin-involved translation examination assay? Though the detail may be described in reference 44, it is difficult for general reader like me to understand the system.

Response: Puromycin is an aminonucleoside antibiotic, which is produced by the bacterium *Streptomyces alboniger*. When cells were treated with proper amounts of puromycin, the nascent polypeptides can be incorporated by puromycin, which reflects directly the rate of global mRNA translation (Schmidt et al, Nat Methods, 2008, PMID: 19305406). Therefore, the Western blot assay using anti-puromycin antibody is an easy-to-implement method to examine translation efficiency in cultured cells. Regarding the two bands, we reason that they are probably non-specific, or puromycin-labelled proteins encoded by house-keeping genes. To make our descriptions much clearer, we have included these method details in the revised section of Materials and Methods.

3. Fig 5J: I do not see a significant reduction of dTAK1 like the case of Fig 6A. Clarify the results in a quantitative manner and explain the possible reasons.

Response: We quantified the protein level of dTak1 in Fig 5J and it was indeed decreased by *eIF3f1 RNAi* (Fig EV4C).

4. Fig 6E & F: I do not see the bands and quantifications of control Flag-Renilla.

Response: Flag-Renilla was utilized as a normalization for the transfection efficiency of expressing plasmids in S2 cells. As it was not altered by *eIF3f1* (Fig 6A), it is therefore not necessary to use Flag-Renilla for other experiments. On the other hand, Tubulin is enough for the quantification of dTak1 protein levels.

5. While this manuscript centers on eIF3f1, there appears to be a lack of comprehensive information concerning the broader role of eIF3s in translation. How does this complex participate in translation? In what ways does this function diverge from other eIFs? What are the sequence similarities and domain compositions when compared to mammalian eIF3s? etc. Incorporating such details, even if the findings extend beyond of translation, it would be valuable and should be included either within the introduction or discussion section.

Response: According to the referee's suggestions, we have discussed in detail regarding the broader roles of eIF3 in the section of Discussion.

6. After having this study, how do author expect the relationship between eIF3f1, dTrabid and Posh? If authors have any data and/or speculations, it will be valuable to describe in the discussion section.

Response: This would be a very open question regarding the regulation of dTak1 ubiquitination/deubiquitination. We would like to speculate two directions for the future studies: (1) a time-course examination of the protein levels of eIF3f1/dTrabid/Posh during IMD signaling (or before/after bacterial infection); (2) investigation of the dynamics in the eIF3f1/dTrabid/Posh/dTak1 interaction network.

7. L104 and L407: But, the data is not Kenny but Relish?

Response: We feel sorry for this mistake and have made correction in the revision.

8. Fig4C: Why is there no comment regarding carbohydrate derivative biosynthetic & metabolic processes?

Response: When the global translation is elevated in the cells with eIF3f1 over-expression, it is reasonable that the global biosynthetic process and metabolic reactions are also enhanced.

9. Fig 4D: Not only top 10 candidates, but also a full list of candidates in the supplemental table will be very much useful to the community.

Response: The full list of candidates has been deposited in the Mendeley Data (<http://data.mendeley.com/datasets/5vnzf8bw3v/draft?a=dbc844ce-2d45-4195-9b3f-bd9e2321a62b>). If our paper is accepted, it will also be published as source data along with Fig 4B.

10. Fig 5C: What could be the reason why eIF3f1 has a lower deubiquitination activity as compared to a positive control vOtu? What could be the determinants of deubiquitination activity? Is this because of the structural difference or sequence conservation?

Response: vOtu harbors well-characterized deubiquitinase enzymatical activities. It is highly active and commonly served as a strong positive control in the *in vitro* deubiquitination assay.

11. L498: μ not u?

Response: We have modified this error accordingly.

12. In materials and methods part, no section for the plasmids?

Response: In the revised version of our manuscript, we have included detailed information of the plasmids.

Referee #3:

In this article, the authors analyze the function in the *Drosophila* host defense against bacterial infection of a putative translation initiation factor, eIF3f1, that contains a JAMM domain thought to mediate deubiquinylation reactions. They first showed both in cultured cells and *in vivo* by RNAi silencing that the loss-of-function phenotype is a reduced induction of antimicrobial peptide (AMP) genes that correlated with a mildly increased sensitivity to a couple of

Gram-negative bacterial infections, consistent with the Immune-deficiency (IMD) pathway being affected, and not the Toll pathway. Next, by relying on overexpression experiments both in cultured cells and in vivo, they showed that the N-terminal JAMM domain is required to mediate an increased IMD-dependent immune response, again the effects being relatively mild. Of note, the overexpression of eIF3f1 rescues the loss-of-function phenotype. A second part of the manuscript deals with the effect on translation and protein stability of eIF3f1 overexpression. This overexpression appeared to increase translation in cultured cells and one of the proteins the level of which is most strongly expressed is the TAK1 kinase, which is a component of the IMD pathway. Interestingly, the increased steady-state levels of dTAK1 are apparently not due to increased protein translation but to decreased proteasome-mediated degradation due to the deubiquitinylation of K48-ubiquitin chains on TAK1 mediated by the JAMM domain. Finally, the effects of eIF3f1 overexpression could be alleviated by silencing TAK1, which suggests that genetically TAK1 functions downstream of eIF3f1.

1. While this study is generally well-conducted, a number of outstanding issues remain. An important claim is that it is the first time that a non-canonical function in immunity is described for a eukaryotic translation initiation factor. While the evidence appears to indeed support a role for this factor in the IMD pathway, although some additional checks are needed (see below), the evidence that eIF3f1 is indeed a translation initiation factor is limited to the overexpression experiment shown in Fig. 4A and implicitly to its homology of sequence to mammalian eIF3f. The authors do not at all mention that an apparent specificity of *Drosophila melanogaster* (and potentially other Dipteran or insect/arthropod species, this reviewer did not take the time to check) is that the genome encodes two paralogs of this factor. eIF3f2 displays 30% identity and some 60% similarity to eIF3f1, which spans rather homogeneously the whole protein. Can the authors exclude that following the duplication of the

locus there has been an evolutionary divergence of function, eIF3f2 being the actual translation initiation factor for instance and eIF3f1 taking on an immune function, possibly among others? The authors should check whether the eIF3f2 factor also contains a JAMM domain and let the reader know whether the mammalian factor also contains this domain. The authors should show that eIF3f1, or eIF3f2, is a bona fide translation initiation factor using a loss-of-function approach as overexpression may be misleading. What happens to TAK1 protein levels when these factors are silenced? A related issue to the existence of the paralog is that it is imperative that the authors experimentally demonstrate that their silencing constructs specifically affect eIF3f1 and not eIF3f2 expression, which may be the case according to VDRC since no computed off-target effect is reported. If the functions of the two paralogs are indeed distinct, the overexpression of eIF3f2 may provide an interesting control.

Response: We are grateful for the referee raising these excellent concerns. To address these questions, we performed a series of additional experiments. We have now included in the revised manuscript the data that the down-regulation of *eIF3f1* promotes endogenous dTak1 ubiquitination and degradation (Figs 5L, M, and EV4D).

Drosophila eIF3f1 and eIF3f2 share 48% and 37% similarities with mammalian EIF3F, respectively (Marygold et al, Fly, 2017, PMID: 27494710). It seems that all three proteins harbor a JAMM domain at the N-terminal region. To investigate the potential involvement of *eIF3f2* in *Drosophila* innate immunity, we performed bacterial infections on both *eIF3f2* RNAi and over-expression flies. We found that neither the down-regulation nor the over-expression of *eIF3f2* affected the fly survival (Fig EV5A) or the AMP expressions (Fig EV5B to D) or the bacterial burden (Fig EV5E) upon microbial infection. In addition, we carried out rescue experiments utilizing *eIF3f2* transgenic fly according to the referee's suggestion. However, ectopic *eIF3f2* didn't rescue the immune defects in the *eIF3f1* RNAi flies (Fig EV5A to E). We

have included these data in the revised version of our manuscript.

2. One point that would enhance the appeal of the second part of this study would be the description of what happens when immune signaling is triggered, especially with respect to TAK1 levels and K48-ubiquitylation in wild-type conditions or when eIF3f1 is either overexpressed or silenced. It would also be worth directly monitoring AMP levels by mass-spectrometry on collected hemolymph (Uttenweiler-Joseph et al, PNAS, 1998) in the different conditions after a mixed Gram-positive and Gram-negative challenge so as to induce both the IMD and the Toll pathway (Drosomycin and Bomanins will serve as internal nonaffected controls, thus bypassing the fact that MALDI-TOF is not especially well-suited for quantitation).

Response: In the revised manuscript, we have included data showing that the K48-linked ubiquitination and degradation of endogenous dTak1 are mediated by eIF3f1 in the *Drosophila* fat body (Fig 5F, G, L, and M). Regarding the examination of AMP expressions, our current data have clearly demonstrated that the down-regulation and the over-expression of *eIF3f1* resulted in, respectively, decreases and increases in the mRNA levels of several AMPs (Figs 1M to O and 2M to O).

Minor points:

1. The title should contain the term *Drosophila*.

Response: According to the referee's suggestion, we have changed the title into "Translation initiation factor eIF3f1 mediates *Drosophila* immune defense in a non-canonical manner".

2. Refs 18 and 19 are not the most relevant when referring to PGRP-LC and-LE. Besides citing the articles that documented their function, the authors ought to cite the structural work that described the direct binding of peptidoglycan fragments to these receptors. The alternative is to cite a review

article.

Response: In the revised manuscript, we have cited the work describing the binding of PGRP receptors to their ligands (Guan et al, Proc Natl Acad Sci USA, 2004, PMID: 15572450).

3. Line 77 and others: "silencing" and not "silence"

Response: We have changed "silence" into "silencing" accordingly.

4. Line 94: Kenny is NOT a kinase, just a key adaptor, which binds to K63 polyubiquitin chains...

Response: We have corrected this issue in the revision.

5. The authors should mention that the c564 driver is not solely expressed in the fat body

Response: We have made modification according to the referee's suggestion.

6. The authors should check whether the melanization and the cellular immune response are affected when eIF3f1 is silenced in flies.

Response: We thank this referee for raising these concerns and deeply agree with the referee that eIF3f1 is potentially involved in melanization and/or cellular immune responses. In the current study, we delve into the function of eIF3f1 in mediating the fly IMD innate immune defense and the underlying molecular mechanism. We would like to focus on this topic in the present study and to demonstrate other potential functions of eIF3f1 in our future projects.

7. In all figures documenting the expression of AMP genes, it is disturbing that the PBS injection controls do not appear to induce their expression, which has been largely documented. The authors should include nonchallenged flies in at least one experiment.

Response: In Fig EV3B in the revision, we have included the data showing the

expression levels of *Dpt* in flies under PBS injection.

8. Line 192: "dramatically"; line 197: "considerable": please, avoid using such hyperbolic terms especially when they do not mirror the data

Response: We have deleted these descriptions in the revised manuscript.

9. Line 211: "indicate" instead of "intimate"

Response: We have changed "intimate" into "indicate" as suggested.

10. FIG. 4D: can the authors exclude that all of their other hits reflect the deubiquitination properties of eIF3f1?

Response: We agree with the referee that this would be an open question. It would be very interesting to test whether eIF3f1 can mediate other proteins in a deubiquitination-dependent manner.

11. The effects of TAK1 silencing appear to be rather mild as compared to those of a null mutant. Can the authors please assess the effectiveness of their RNAi line and mention the limitation of RNAi, the risk of generating hypomorphic phenotypes?

Response: This *dTak1 RNAi* strain was obtained from the Tsinghua RNAi Center and worked quite well in previous studies (Cai et al, *iScience*, 2021, PMID: 33665569; Zhu et al, *Aging and Disease*, 2021, PMID: 34631223).

12. Statistics: the type of statistics used should be described for each panel (ANOVA or Mann-Whitney, and why they use it). Actually, except for Fig. 4F and 5K, the authors perform multiple comparisons: they should use the Kruskal-Wallis test for non-parametric tests and also perform post-hoc tests for all multiple comparison tests).

Response: We have performed additional statistical analyses according to the referee's suggestions. In addition, we have included detailed information regarding statistics in each panel.

Dear Prof. Ji,

Thank you for the submission of your revised manuscript to our editorial offices. I have now received the reports from the three referees that I asked to re-evaluate your study, you will find below. As you will see, the referees support the publication of the study in EMBO reports, but they all have remaining concerns and suggestions to improve the manuscript, I ask you to address in a final revised manuscript. Please also provide a detailed final p-b-p-response regarding these points.

Moreover, I have these editorial requests I ask you to address in a final revised manuscript:

- Please provide individual production quality figure files as .eps, .tif, .jpg (one file per figure) of the five EV figures. Please upload these as separate, individual files upon re-submission.

- Please reduce the number of keywords to 5 and order the manuscript sections like this, using these names:

Title page - Abstract - Keywords - Introduction - Results - Discussion - Materials and Methods - Data availability section - Acknowledgements - Disclosure and Competing Interests Statement - References - Figure legends - Expanded View Figure legends

- The Data Availability section should only contain information on large datasets that have been deposited to external repositories and all access information. Please remove the sentence 'All data needed to evaluate the conclusions are present in the paper' and provide information on the mass spectrometry data.

We require that primary datasets produced (e.g. RNA-seq, ChIP-seq, mass spec., structural and array data) are deposited in an appropriate public database. See also: <http://embor.embopress.org/authorguide#datadeposition>

The accession numbers and database should be listed in the "Data Availability" section (placed after Materials & Methods) following the model below. Please note that the Data Availability Section is restricted to new primary data that are part of this study.

Data availability

- Please make sure that the number "n" for how many independent experiments were performed, their nature (biological versus technical replicates), the bars and error bars (e.g. SEM, SD) and the test used to calculate p-values is indicated in the respective figure legends (for main and EV figures) of the final revised manuscript. Please also check that all the p-values are explained in the legend, and that these fit to those shown in the figure. Please provide statistical testing where applicable. Please avoid the phrase "independent experiment", but clearly state if these were biological or technical replicates. Please also indicate (e.g. with n.s.) if testing was performed, but the differences are not significant. In case n=2, please show the data as separate datapoints without error bars and statistics. See also:

<http://www.embopress.org/page/journal/14693178/authorguide#statisticalanalysis>

If n<5, please show single datapoints for diagrams. Presently, it seems that information related to n is missing in the legend of figures 1a-h, k-q; 2b-i, k-q; 3b-h; 4h; 5e, g, i, k, m; 6b, d, f, h, i-k, m-p; EV1a-f; EV2a-f; EV3a-e; EV4b-d; EV5a-e. Please check.

- Please add to each legend (for main and EV figures) a 'Data Information' section explaining the statistics used or providing information regarding replicates and scales.

- Please provide the reference list in our reference format:

- Please add Table 1 to the end of the main manuscript text file (as word table) and add a title and a legend.

- Please note that all corresponding authors are required to supply an ORCID ID for their name upon submission of a revised manuscript. Please do that for co-corresponding author Qingshuang Cai. Please find instructions on how to link the ORCID ID to the account in our manuscript tracking system in our Author guidelines:

In addition, I would need from you:

Yours sincerely,

Referee #1:

In this revised manuscript, most of the critiques from the first submission have been thoroughly addressed, often with new data. Two points, however, need to be addressed:

1. In figure 5L and M, something is amiss, or I do not understand the graph in panel M. Panel L clearly shows increased K48 conjugated TAK1 on Eif2f1 RNAi, with a similar amount of TAK1 IPed (even though the total lysate TAK1 is lower, as predicted). Yet the graph shows lower K48(IP)/TAK(IP) ratios. Either something is mislabelled, or ratios are flipped, or I don't understand something here. Please correct and/or find better way to display the quantization of this results (more K48Ub in the KD of Eif2f1).
2. The response to the critique of the MG132 experiment, upper part of Figure 6 is inadequate. These blots (the upper reaches thereof) should be probed for K48Ub, and it should be markedly increased with MG132 as this should cause the accumulation of the Ub conjugated intermediate. Ideally, this would be shown with endogenous TAK1 as well as transfected.

Referee #2:

Major points:

1. It is still unclear how and why the authors initially chose to concentrate on eIF3f1 (L83-L87 and L99-L100): In references of 39 and 40, it is noted that not only eIF3f1 but also eIF3f2 and eIF3fh possess a JAMM domain. This raises at least two questions: 1) the rationale behind the authors' immediate focus on investigating the function of eIF3f1, and 2) the roles of other eIF3 subunits in the activation of the IMD pathway.

Response: We initiated this work based on a genetic screening in S2 cells utilizing the Att-Luc reporter system. In this screening, we identified eIF3f1 as a potential modulator in the IMD signalling pathway. We further conducted both in vitro and in vivo investigations to address the functional assessment of eIF3f1 in the fly antimicrobial immune defense and the underlying molecular mechanism.

I appreciate the author's information; however, it is still unclear which candidate genes were selected for the initial genetic screen in S2 cells. How did this screen lead to the identification of eIF3f1, and what was the reason to pursue the function of eIF3f1? I understand that this manuscript focuses on the function of eIF3f1; however, it is still awkward to me that, from the first sentence of the Results section (Line 102), the authors immediately delve into the direct functional analyses of eIF3f1 without providing any information about the introductory screen.

Regarding the potential contribution of eIF3f2 to regulating the innate immune defense in *Drosophila* (also raised by referee#1), we have performed additional experiments showing that neither the down-regulation nor the over-expression of eIF3f2 affects the fly innate immune reactions upon bacterial infections (FigEV5A to E in the revision). To our best knowledge, it has remained largely unknown whether other eIF3 subunits are involved in modulating the IMD signalling pathway. We definitely agree with the referee and would like to explore the potential functional roles of other eIF3 components in *Drosophila* innate immunity in our future projects.

OK and a small correction from my first comment. Among a total of 16 eIF3 subunits (based on the flybase), three namely eIF3f1, eIF3f2 and eIF3h (not fh) have a JAMM domain.

2. Regarding the aforementioned point: Based on the publication by Moretti et al in PLoS Biol (2010) (PMID21124883), it is noteworthy that mammalian eIF3h also contains a JAMM domain, and its intrinsic deubiquitinase activity regulates the Notch signalling pathway. While the report did not explore its involvement in innate immune signalling pathways, it seems that this manuscript might not be the first example to identify a role for a translation initiation factor at the level of post-translational regulation. Please clarify this point and if authors agree with this, tone down their claim.

Response: We are grateful for the referee pointing out these issues. In the revised manuscript, we have cited this paper and toned down our claims.

OK

3. Previous RNAi studies conducted in S2 cells have revealed that numerous genes implicated in translation exhibit impairments in cell viability. Even if the results are based on the normalization of either Actin-Renilla activity or Rpl32 expression, I am curious about the potential impact of eIF3f1 knockdown and overexpression on these normalization controls. If such an influence exists, it would be important to incorporate these observations within the manuscript and make necessary adjustments to certain interpretations.

Response: In fact, we didn't observe marked alterations regarding Renilla activities and Rpl32 expression levels, when we performed dual-luciferase and RT-qPCR assays under the conditions of eIF3f1 RNAi or over-expression.

OK

Minor points:

1. Fig 1: Though authors elegantly showed their confirmation in vivo in Fig 2, The data of Fig 1 are all based in the context of IMD overexpression. How about the phenotype on the physiological heat-killed E.coli stimulated condition? Plus, in Fig 2E-H, there is no data point of simple eIF3f1 overexpression.

Response: To address this concern (also raised by referee#1), we have included in the revised manuscript the data using E. coli stimulation (FigEV1A to F). In addition, we performed experiments examining the relative luciferase activities with only eIF3f1 over-expression. These results have been shown in FigEV1D to F in the revision.

OK

2. Fig 4A and L198-L199: What are the two bands (~65kDa and ~40 kDa) corresponding to? What is the experimental system of using puromycin-involved translation examination assay? Though the detail may be described in reference 44, it is difficult for general reader like me to understand the system.

Response: Puromycin is an aminonucleoside antibiotic, which is produced by the bacterium *Streptomyces alboniger*. When cells were treated with proper amounts of puromycin, the nascent polypeptides can be incorporated by puromycin, which reflects directly the rate of global mRNA translation (Schmidt et al, Nat Methods, 2008, PMID: 19305406). Therefore, the Western blot assay using anti-puromycin antibody is an easy-to-implement method to examine translation efficiency in cultured cells. Regarding the two bands, we reason that they are probably non-specific, or puromycin-labelled proteins encoded by house-keeping genes. To make our descriptions much clearer, we have included these method details in the revised section of Materials and Methods.

OK

3. Fig 5J: I do not see a significant reduction of dTAK1 like the case of Fig 6A. Clarify the results in a quantitative manner and explain the possible reasons.

Response: We quantified the protein level of dTak1 in Fig 5J and it was indeed decreased by eIF3f1 RNAi (FigEV4C).

OK

4. Fig 6E & F: I do not see the bands and quantifications of control Flag-Renilla.

Response: Flag-Renilla was utilized as a normalization for the transfection efficiency of expressing plasmids in S2 cells. As it was not altered by eIF3f1 (Fig 6A), it is therefore not necessary to use Flag-Renilla for other experiments. On the other hand,

Tubulin is enough for the quantification of dTak1 protein levels.

OK

5. While this manuscript centers on eIF3f1, there appears to be a lack of comprehensive information concerning the broader role of eIF3s in translation. How does this complex participate in translation? In what ways does this function diverge from other eIFs? What are the sequence similarities and domain compositions when compared to mammalian eIF3s? etc. Incorporating such details, even if the findings extend beyond of translation, it would be valuable and should be included either within the introduction or discussion section.

Response: According to the referee's suggestions, we have discussed in detail regarding the broader roles of eIF3 in the section of Discussion.

OK

6. After having this study, how do author expect the relationship between eIF3f1, dTrabid and Posh? If authors have any data and/or speculations, it will be valuable to describe in the discussion section.

Response: This would be a very open question regarding the regulation of dTak1 ubiquitination/deubiquitination. We would like to speculate two directions for the future studies: (1) a time-course examination of the protein levels of eIF3f1/dTrabid/Posh during IMD signalling (or before/after bacterial infection); (2) investigation of the dynamics in the eIF3f1/dTrabid/Posh/dTak1 interaction network.

Since it is nice directions, please mention these in the discussion part.

7. L104 and L407: But, the data is not Kenny but Relish?

Response: We feel sorry for this mistake and have made correction in the revision.

OK

8. Fig4C: Why is there no comment regarding carbohydrate derivative biosynthetic & metabolic processes?

Response: When the global translation is elevated in the cells with eIF3f1 over-expression, it is reasonable that the global biosynthetic process and metabolic reactions are also enhanced.

OK

9. Fig 4D: Not only top 10 candidates, but also a full list of candidates in the supplemental table will be very much useful to the community.

Response: The full list of candidates has been deposited in the Mendeley Data (<http://data.mendeley.com/datasets/5vznzf8bw3v/draft?a=dbc844ce-2d45-4195-9b3f-bd9e2321a62b>). If our paper is accepted, it will also be published as source data along with Fig 4B.

OK

10. Fig 5C: What could be the reason why eIF3f1 has a lower deubiquitination activity as compared to a positive control vOtu? What could be the determinants of deubiquitination activity? Is this because of the structural difference or sequence conservation?

Response: vOtu harbors well-characterized deubiquitinase enzymatical activities. It is highly active and commonly served as a strong positive control in the in vitro deubiquitination assay.

Okay, but my question is: What characteristics determine the deubiquitinase enzymatic activity? How do the catalytic residues differ between vOtu and eIF3f1? Because this assay is purely based on in vitro examination, I expect that the difference should not be attributed to expression levels, distinct tissue localization, or different interacting partners.

11. L498: μ not u ?

Response: We have modified this error accordingly.

OK

12. In materials and methods part, no section for the plasmids?

Response: In the revised version of our manuscript, we have included detailed information of the plasmids.

OK

Additional minor points:

1. L895: what is MOI (how many bacteria per cell)?
2. Why did the authors not test the phenotype of eiF3f2 in the S2 RNAi system? This is generally faster to test. Although the results from in vivo experiments are much more valuable, the findings showed no effect. Thus, at least two possibilities remain: 1) eiF3f2 RNAi did not work well, or 2) the RNAi itself worked, but it was not strong enough to demonstrate an AMP phenotype.

Referee #3:

This reviewer acknowledges that several concerns have been successfully addressed.

However, one especially important point that is central to this manuscript has not been rigorously addressed, a disappointment. Namely, the authors did not experimentally demonstrate that either eiF3f1, eiF3f2, or BOTH OF THEM TOGETHER are REQUIRED for translation (overexpression studies alone are not sufficient). As stated earlier (middle of point 1) but unfortunately not noticed, the biological question is whether one of the duplicated genes has evolved to take on an immune role - the data are compatible with this possibility- whereas the other may or may not have fully taken on the translation initiation function. HAS ANYONE ACTUALLY SHOWN THAT THESE FACTORS WORK AS TRANSLATION INITIATION FACTORS IN Drosophila? If only eiF3f2 is playing a role in translation initiation, then the title is not correct. Given the data available in Flybase and other resources, it should be a Master student's game to determine whether the duplication is shared between protostomes and thus occurred relatively early during evolution or is a specialization of insects.

The RNAi lines used against eiF3f2 have not been validated and no proof is provided that they successfully silence the expression of their target. Thus, the absence of a susceptibility to infection phenotype is at best indicative unless the author can show that the RNAi line is blocking other functions of eiF3f2, e.g., translation.

Minor points:

Answer to Point 2: if the reviewer is asking for MS data and not transcriptional data as regards the actual expression of Antimicrobial PEPTIDES, there must be a good reason to perform this control in an article dealing with translation initiation factors (it may actually be a good way to discriminate between the respective functions of the two eiF3f genes: these peptides are massively produced).

Writing:

- a. please, pay attention not to fall in the biological finalism trap, a mistake we all make at one point or the other. "to" being used as a shortened form of "in order to": lines 28,94, 257: for instance, the association of eiF3f1 (better scratch out the adjective physical or alternatively mention a TAK1-containing complex) with TAK1 results in the modulation of Tak1 turnover; there is no "great architect" designing this association, just random variations that are evolutionarily selected.
- b. Overstatements: please, delete the adjectives "crucial" line 136, "essential" line 165; line 290 may-be replace "governs" that is likely too strong, possibly by "modulates".
- c. Line 135 Drosomycin and Metchnikowin (initial capital letter as per FlyBase)
- d. "Gram" not gram, please: Prof. Hans Christian Gram invented this highly popular bacterial staining method.

Statistics: the authors are to be commended for using the Kruskal-Wallis test instead of Mann-Whitney when multiple comparisons are performed. They fail however to indicate which post-hoc test was performed when subsequently comparing data sets two by two (Tukey?).

Referee #1:

In this revised manuscript, most of the critiques from the first submission have been thoroughly addressed, often with new data. Two points, however, need to be addressed:

1. In figure 5L and M, something is amiss, or I do not understand the graph in panel M. Panel L clearly shows increased K48 conjugated TAK1 on Eif2f1 RNAi, with a similar amount of TAK1 IPed (even though the total lysate TAK1 is lower, as predicted). Yet the graph shows lower K48(IP)/TAK(IP) ratios. Either something is mislabelled, or ratios are flipped, or I don't understand something here. Please correct and/or find better way to display the quantization of this results (more K48Ub in the KD of Eif2f1).

Response: We deeply appreciate that the referee pointed out this mistake. In the revised manuscript, we have included the correct Fig 5M quantifying the K48-linked ubiquitination level of dTak1 under different conditions.

2. The response to the critique of the MG132 experiment, upper part of Figure 6 is inadequate. These blots (the upper reaches thereof) should be probed for K48Ub, and it should be markedly increased with MG132 as this should cause the accumulation of the Ub conjugated intermediate. Ideally, this would be shown with endogenous TAK1 as well as transfected.

Response: According to the referee's suggestion, we performed Western blot assays using anti-K48Ub and anti-dTak1 antibodies. As shown in Fig EV4E in the revision, MG132 treatment led to accumulated K48-linked ubiquitination in S2 cells.

Referee #2:

Major points:

1. It is still unclear how and why the authors initially chose to concentrate on eIF3f1 (L83-L87 and L99-L100): In references of 39 and 40, it is noted that not only eIF3f1 but also eIF3f2 and eIF3fh possess a JAMM domain. This raises at least two questions: 1) the rationale behind the authors' immediate focus on investigating the function of eIF3f1, and 2) the roles of other eIF3 subunits in the activation of the IMD pathway.

Response: We initiated this work based on a genetic screening in S2 cells utilizing the Att-Luc reporter system. In this screening, we identified eIF3f1 as a potential modulator in the IMD signaling pathway. We further conducted both in vitro and in vivo investigations to address the functional assessment of eIF3f1 in the fly antimicrobial immune defense and the underlying molecular mechanism.

I appreciate the author's information; however, it is still unclear which candidate genes were selected for the initial genetic screen in S2 cells. How did this screen lead to the identification of eIF3f1, and what was the reason to pursue the function of eIF3f1? I understand that this manuscript focuses on the function of eIF3f1; however, it is still awkward to me that, from the first sentence of the Results section (Line 102), the authors immediately delve into the direct functional analyses of eIF3f1 without providing any information about the introductory screen.

Response: In fact, we performed genetic screenings unbiasedly on hundreds of

genes from an arrayed dsRNA library. When we first observed that *eIF3f1 RNAi* showed a phenotype, we of course would like to set up further investigations to verify its potential function in regulating *Drosophila* innate immunity and the underlying molecular mechanism. We have to admit that we were quite lucky to find *eIF3f1*. Nevertheless, similar cases happened at times in the long history of human's scientific researches and we respectfully think they were not awkward.

Regarding the potential contribution of eIF3f2 to regulating the innate immune defense in *Drosophila* (also raised by referee#1), we have performed additional experiments showing that neither the down-regulation nor the over-expression of eIF3f2 affects the fly innate immune reactions upon bacterial infections (FigEV5A to E in the revision). To our best knowledge, it has remained largely unknown whether other eIF3 subunits are involved in modulating the IMD signalling pathway. We definitely agree with the referee and would like to explore the potential functional roles of other eIF3 components in *Drosophila* innate immunity in our future projects.

OK and a small correction from my first comment. Among a total of 16 eIF3 subunits (based on the flybase), three namely eIF3f1, eIF3f2 and eIF3h (not fh) have a JAMM domain.

Response: Thanks.

2. Regarding the aforementioned point: Based on the publication by Moretti et al in PLoS biol (2010) (PMID21124883), it is noteworthy that mammalian eIF3h also

contains a JAMM domain, and its intrinsic deubiquitinase activity regulates the Notch signalling pathway. While the report did not explore its involvement in innate immune signalling pathways, it seems that this manuscript might not be the first example to identify a role for a translation initiation factor at the level of post-translational regulation. Please clarify this point and if authors agree with this, tone down their claim.

Response: We are grateful for the referee pointing out these issues. In the revised manuscript, we have cited this paper and toned down our claims.

OK

Response: Thanks.

3. Previous RNAi studies conducted in S2 cells have revealed that numerous genes implicated in translation exhibit impairments in cell viability. Even if the results are based on the normalization of either Actin-Renilla activity or RpL32 expression, I am curious about the potential impact of eIF3f1 knockdown and overexpression on these normalization controls. If such an influence exists, it would be important to incorporate these observations within the manuscript and make necessary adjustments to certain interpretations.

Response: In fact, we didn't observe marked alterations regarding Renilla activities and RpL32 expression levels, when we performed dual-luciferase and RT-qPCR assays under the conditions of eIF3f1RNAi or over-expression.

OK

Response: Thanks.

Minor points:

1. Fig 1: Though authors elegantly showed their confirmation in vivo in Fig 2, The data of Fig 1 are all based in the context of IMD overexpression. How about the phenotype on the physiological heat-killed E.coli stimulated condition? Plus, in Fig 2E-H, there is no data point of simple eIF3f1 overexpression.

Response: To address this concern (also raised by referee#1), we have included in the revised manuscript the data using E. coli stimulation (FigEV1A to F). In addition, we performed experiments examining the relative luciferase activities with only eIF3f1 over-expression. These results have been shown in FigEV1D to F in the revision.

OK

Response: Thanks.

2. Fig 4A and L198-L199: What are the two bands (~65kDa and ~40 kDa) corresponding to? What is the experimental system of using puromycin-involved translation examination assay? Though the detail may be described in reference 44, it is difficult for general reader like me to understand the system.

Response: Puromycin is an aminonucleoside antibiotic, which is produced by the bacterium *Streptomyces alboniger*. When cells were treated with proper amounts of puromycin, the nascent polypeptides can be incorporated by puromycin, which

reflects directly the rate of global mRNA translation (Schmidt et al, Nat Methods, 2008, PMID: 19305406). Therefore, the Western blot assay using anti-puromycin antibody is an easy-to-implement method to examine translation efficiency in cultured cells. Regarding the two bands, we reason that they are probably non-specific, or puromycin-labelled proteins encoded by house-keeping genes. To make our descriptions much clearer, we have included these method details in the revised section of Materials and Methods.

OK

Response: Thanks.

3. Fig 5J: I do not see a significant reduction of dTAK1 like the case of Fig 6A. Clarify the results in a quantitative manner and explain the possible reasons.

Response: We quantified the protein level of dTak1 in Fig 5J and it was indeed decreased by eIF3f1 RNAi (FigEV4C).

OK

Response: Thanks.

4. Fig 6E & F: I do not see the bands and quantifications of control Flag-Renilla.

Response: Flag-Renilla was utilized as a normalization for the transfection efficiency of expressing plasmids in S2 cells. As it was not altered by eIF3f1(Fig 6A), it is therefore not necessary to use Flag-Renilla for other experiments. On the other hand, Tubulin is enough for the quantification of dTak1protein levels.

OK

Response: Thanks.

5. While this manuscript centers on eIF3f1, there appears to be a lack of comprehensive information concerning the broader role of eIF3s in translation. How does this complex participate in translation? In what ways does this function diverge from other eIFs? What are the sequence similarities and domain compositions when compared to mammalian eIF3s? etc. Incorporating such details, even if the findings extend beyond of translation, it would be valuable and should be included either within the introduction or discussion section.

Response: According to the referee's suggestions, we have discussed in detail regarding the broader roles of eIF3 in the section of Discussion.

OK

Response: Thanks.

6. After having this study, how do author expect the relationship between eIF3f1, dTrabid and Posh? If authors have any data and/or speculations, it will be valuable to describe in the discussion section.

Response: This would be a very open question regarding the regulation of dTak1 ubiquitination/deubiquitination. We would like to speculate two directions for the future studies: (1) a time-course examination of the protein levels of eIF3f1/dTrabid/Posh during IMD signalling (or before/after bacterial infection); (2) investigation of the

dynamics in the eIF3f1/dTrabid/Posh/dTak1 interaction network.

Since it is nice directions, please mention these in the discussion part.

Response: We have included this part in the Discussion section in the revised manuscript.

7. L104 and L407: But, the data is not Kenny but Relish?

Response: We feel sorry for this mistake and have made correction in the revision.

OK

Response: Thanks.

8. Fig4C: Why is there no comment regarding carbohydrate derivative biosynthetic & metabolic processes?

Response: When the global translation is elevated in the cells with eIF3f1 over-expression, it is reasonable that the global biosynthetic process and metabolic reactions are also enhanced.

OK

Response: Thanks.

9. Fig 4D: Not only top 10 candidates, but also a full list of candidates in the supplemental table will be very much useful to the community.

Response: The full list of candidates has been deposited in the Mendeley Data (<http://data.mendeley.com/datasets/5vnzf8bw3v/draft?a=dbc844ce-2d45-4195-9b3f->

bd9e2321a62b). If our paper is accepted, it will also be published as source data along with Fig 4B.

OK

Response: Thanks.

10. Fig 5C: What could be the reason why eIF3f1 has a lower deubiquitination activity as compared to a positive control vOtu? What could be the determinants of deubiquitination activity? Is this because of the structural difference or sequence conservation?

Response: vOtu harbors well-characterized deubiquitinase enzymatical activities. It is highly active and commonly served as a strong positive control in the *in vitro* deubiquitination assay.

Okay, but my question is: What characteristics determine the deubiquitinase enzymatic activity? How do the catalytic residues differ between vOtu and eIF3f1? Because this assay is purely based on *in vitro* examination, I expect that the difference should not be attributed to expression levels, distinct tissue localization, or different interacting partners.

Response: We agree with the referee that these experiments were performed *in vitro*, and the amount of protein used in the assays is one of the major factors affecting the results. For this, we have clearly stated in the section of Materials and Methods that 100 ng of each protein was used in the *in vitro* Dub assays. On the other hand, deubiquitinase enzymes may execute their enzymatical activities through distinct

catalytic residues or domains. For the cases of vOtu and eIF3f1, their catalytic domains are OTU and JAMM, respectively. However, we cannot obtain evident information regarding the differences of these domains, nor their structures, localizations, interacting partners, based on the results of *in vitro* Dub assays.

11. L498: μ not u?

Response: We have modified this error accordingly.

OK

Response: Thanks.

12. In materials and methods part, no section for the plasmids?

Response: In the revised version of our manuscript, we have included detailed information of the plasmids.

OK

Response: Thanks.

Additional minor points:

1. L895: what is MOI (how many bacteria per cell)?

Response: The MOI is around 5 bacteria per S2 cell. We have included this piece of information in L934 in the revision.

2. Why did the authors not test the phenotype of eIF3f2 in the S2 RNAi system? This

is generally faster to test. Although the results from *in vivo* experiments are much more valuable, the findings showed no effect. Thus, at least two possibilities remain: 1) *eIF3f2* RNAi did not work well, or 2) the RNAi itself worked, but it was not strong enough to demonstrate an AMP phenotype.

Response: We deeply agree with the referee that *in vivo* experiments are much more valuable. In this regard, there is no need to examine the functional role of *eIF3f2* *in vitro*.

Referee #3:

This reviewer acknowledges that several concerns have been successfully addressed. However, one especially important point that is central to this manuscript has not been rigorously addressed, a disappointment. Namely, the authors did not experimentally demonstrate that either *eiF3f1*, *eiF3f2*, or BOTH OF THEM TOGETHER are REQUIRED for translation (overexpression studies alone are not sufficient). As stated earlier (middle of point 1) but unfortunately not noticed, the biological question is whether one of the duplicated genes has evolved to take on an immune role -the data are compatible with this possibility- whereas the other may or may not have fully taken on the translation initiation function. HAS ANYONE ACTUALLY SHOWN THAT THESE FACTORS WORK AS TRANSLATION INITIATION FACTORS IN *Drosophila*? If only *eiF3f2* is playing a role in translation initiation, then the title is not correct. Given the data available in Flybase and other resources, it should be a Master student's game to determine whether the duplication

is shared between protostomes and thus occurred relatively early during evolution or is a specialization of insects.

Response: In this work, we delve into investigating the functional involvement of eIF3f1 in the fly antimicrobial immune defense and the underlying molecular mechanism. For the part of mechanism study, we have provided compelling evidence that *Drosophila* eIF3f1 modulates the proteasome-involved degradation, but not the translation of dTak1. We definitely agree with the referee that the functional role of *Drosophila* eIF3f1 in translation initiation has remained elusive. Regardless, our investigations in S2 cells showed that over-expression of eIF3f1 enhanced the global translation. In our future projects, we would like to systematically examine the role of eIF3f1 in translation initiation and the related biological functions. According to the referee's suggestions, we have changed the Title into "*Drosophila* eIF3f1 mediates the host immune defense through targeting dTak1". In addition, we have included these pieces of information in the Discussion section.

The RNAi lines used against eIF3f2 have not been validated and no proof is provided that they successfully silence the expression of their target. Thus, the absence of a susceptibility to infection phenotype is at best indicative unless the author can show that the RNAi line is blocking other functions of eIF3f2, e.g., translation.

Response: In Fig EV5A of the revised manuscript, we have included the validation data showing the RNAi efficiency of *eIF3f2*. As *eIF3f2* didn't show a phenotype in our experimental system, it would be more appropriate to access its potential functions in

our future projects.

Minor points:

Answer to Point 2: if the reviewer is asking for MS data and not transcriptional data as regards the actual expression of Antimicrobial PEPTIDES, there must be a good reason to perform this control in an article dealing with translation initiation factors (it may actually be a good way to discriminate between the respective functions of the two eIF3f genes: these peptides are massively produced).

Response: We are grateful for the referee providing such constructive suggestions. As we mentioned above, we found that eIF3f1 regulates the ubiquitination and proteasome-involved turnover of dTak1, thereby mediating the fly antimicrobial immune defense. Our current data cannot exclude other potential mechanisms (e.g. translational regulation of AMPs), by which eIF3f1 execute its immune function. To keep the rigor of our study, we have modified the Title and included these pieces information in the Discussion section.

Writing:

a. please, pay attention not to fall in the biological finalism trap, a mistake we all make at one point or the other. "to" being used as a shortened form of "in order to": lines 28,94, 257: for instance, the association of eIF3f1 (better scratch out the adjective physical or alternatively mention a TAK1-containing complex) with TAK1 results in the modulation of Tak1 turnover; there is no "great architect" designing this association,

just random variations that are evolutionarily selected.

Response: We have changed “to” into “and” in the revised text.

b. Overstatements: please, delete the adjectives "crucial" line 136, "essential" line 165; line 290 may-be replace "governs" that is likely too strong, possibly by "modulates".

Response: We have made modifications as suggested.

c. Line 135 Drosomycin and Metchnikowin (initial capital letter as per FlyBase)

Response: We have corrected them.

d. "Gram" not gram, please: Prof. Hans Christian Gram invented this highly popular bacterial staining method.

Response: We have revised this in the revision.

Statistics: the authors are to be commended for using the Kruskal-Wallis test instead of Mann-Whitney when multiple comparisons are performed. They fail however to indicate which post-hoc test was performed when subsequently comparing data sets two by two (Tukey?).

Response: The Tukey’s test was used for statistical analysis between two groups. We have included this piece of information in the section of Materials and Methods.

Dear Prof. Ji,

Thank you for the submission of your further revised manuscript to our editorial offices. I have now received the reports from the three referees that I have asked to look into this again, you will find below. As you will see, the referees now support the publication of your study in EMBO reports. Nevertheless, referee #3 has some final suggestions to improve the manuscript and the title, I ask you to address in a final revised manuscript.

During cross-commenting, all three referees also indicated that it is absolutely necessary to describe clearly how the initial screen was performed and why eIF3f1 was picked up from other hits. For this, we also need a paragraph in the methods section describing the screening procedure.

Yours sincerely,

Referee #1:

The revised manuscript address all prior critiques from this reviewer.

I am not sure Reviewer #3 will be satisfied, but I am not sure I agree with his comment, that the authors need to, in this work, figure out how these two Eif3f is-forms control translation. It would be nice, but I would agree not needed here for the conclusions they want to make. Their conclusions are sound and interesting!

Referee #2:

The authors have addressed all of this reviewer's comments. Except for a small mistake that there is a redundant reference between L643 and L700, I have no further comments.

Referee #3:

This reviewer regrets that the authors did not experimentally address the issue as to whether eIF4f1 is a true translation factor. As a consequence, it is imperative that they rephrase some of their statements as detailed below, starting with the title that is not accurate: the authors cannot claim that eIF3f1 is a translation factor. The current title is therefore inappropriate and potentially misleading.

Last paragraph of the Introduction: the authors should already state there that there are two Drosophila homologs of mammalian translation initiation factor eIF3F and that they focus in this article on eIF3f1.

Line 199: "As eIF3f1 is POTENTIALLY an integral component..."

Line 297 (Discussion): this sentence cannot remain as such as there is no proof that eIF3f1 is a translation factor. At best the authors could mention the homology to mammalian eIF3F.

Line 314: " we speculated that as POTENTIALLY part of the translational initiation factors"

Line 322: same problem as for line 297.

Paragraph lines 325-334: the authors should state explicitly that a translation function for eIF3f1 is not demonstrated and that it is not possible to formally exclude the possibility that the translation initiation function might be performed by eIF3f2.

Referee #1:

The revised manuscript address all prior critiques from this reviewer.

I am not sure Reviewer #3 will be satisfied, but I am not sure I agree with his comment, that the authors need to, in this work, figure out how these two Eif3f is-forms control translation. It would be nice, but I would agree not needed here for the conclusions they want to make. Their conclusions are sound and interesting!

Response: We are grateful for the positive feedback from this referee.

Referee #2:

The authors have addressed all of this reviewer's comments. Except for a small mistake that there is a redundant reference between L643 and L700, I have no further comments.

Response: Thanks. We have revised the references.

Referee #3:

This reviewer regrets that the authors did not experimentally address the issue as to whether eIF4f1 is a true translation factor. As a consequence, it is imperative that they rephrase some of their statements as detailed below, starting with the title that is not accurate: the authors cannot claim that eIF3f1 is a translation factor. The current title is therefore inappropriate and potentially misleading.

Response: As we have revised the title of our manuscript into "*Drosophila* eIF3f1 mediates the host immune defense through targeting dTak1", we politely think this title

cannot mislead the readers.

Last paragraph of the Introduction: the authors should already state there that there are two *Drosophila* homologs of mammalian translation initiation factor eIF3F and that they focus in this article on eIF3f1.

Response: We have included this piece of information in the revised manuscript.

Line 199: "As eIF3f1 is POTENTIALLY an integral component..."

Response: Revised.

Line 297 (Discussion): this sentence cannot remain as such as there is no proof that eIF3f1 is a translation factor. At best the authors could mention the homology to mammalian eIF3F.

Response: Revised.

Line 314: " we speculated that as POTENTIALLY part of the translational initiation factors"

Response: Revised.

Line 322: same problem as for line 297.

Response: Revised.

Paragraph lines 325-334: the authors should state explicitly that a translation function for eIF3f1 is not demonstrated and that it is not possible to formally exclude the possibility that the translation initiation function might be performed by eIF3f2.

Response: We have stated in the Discussion section that “the functional involvement of *Drosophila* eIF3f1 in translation initiation needs to be further investigated in more details”. Regarding eIF3f2, we have included the description that “Regardless, it is not possible to formally exclude the possibility that the translation initiation function might be performed by eIF3f2”.

Prof. Shanming ji
Anhui Agricultural University
Changjiang West Road
Anhui 230036
China

Dear Prof. ji,

I am very pleased to accept your manuscript for publication in the next available issue of EMBO reports. Thank you for your contribution to our journal.

Yours sincerely,
